# A zebrafish and mouse model for selective pruritus via direct activation of TRPA1

Kali Esancy[1,2†], Logan Condon[1†], Jing Feng[3†], Corinna Kimball[1], Andrew Curtright[1], Ajay Dhaka[1,2*]

[1]Department of Biological Structure, University of Washington, Seattle, United States; [2]Graduate Program in Neuroscience, University of Washington, Seattle, United States; [3]Center for the Study of Itch, Washington University, St. Louis, United States

**Abstract** Little is known about the capacity of lower vertebrates to experience itch. A screen of itch-inducing compounds (pruritogens) in zebrafish larvae yielded a single pruritogen, the TLR7 agonist imiquimod, that elicited a somatosensory neuron response. Imiquimod induced itch-like behaviors in zebrafish distinct from those induced by the noxious TRPA1 agonist, allyl isothiocyanate. In the zebrafish, imiquimod-evoked somatosensory neuronal responses and behaviors were entirely dependent upon TRPA1, while in the mouse TRPA1 was required for the direct activation of somatosensory neurons and partially responsible for behaviors elicited by this pruritogen. Imiquimod was found to be a direct but weak TRPA1 agonist that activated a subset of TRPA1 expressing neurons. Imiquimod-responsive TRPA1 expressing neurons were significantly more sensitive to noxious stimuli than other TRPA1 expressing neurons. Together, these results suggest a model for selective itch via activation of a specialized subpopulation of somatosensory neurons with a heightened sensitivity to noxious stimuli.

DOI: https://doi.org/10.7554/eLife.32036.001

*For correspondence: dhaka@uw.edu

†These authors contributed equally to this work

Competing interests: The authors declare that no competing interests exist.

## Introduction

Itch is an unpleasant sensation that elicits a scratch behavior in terrestrial vertebrates. In mammals, chemically-induced itch is thought to be mediated by pruritic receptors on somatosensory neurons (*Bautista et al., 2014*; *Hoon, 2015*). These receptors are typically G-protein coupled receptors (GPCRs) that, upon activation, prompt the opening of downstream transient receptor potential (TRP) channels, facilitating activation of the neuron (*Ross, 2011*; *Zhang, 2015*). This coupling of pruritic receptors to TRPA1 or TRPV1 is especially intriguing in that these TRP channels also serve as nociceptors, mediating responses to algogenic (painful) stimuli (*Laing and Dhaka, 2016*; *Ikoma et al., 2006*).

Zebrafish (*Danio rerio*) have proven to be a valuable tool in the study of nociception (*Gau et al., 2013*). The zebrafish ortholog of *Trpa1*, *trpa1b*, is required for nociceptive responses to aversive pungent chemicals (*Prober et al., 2008*). Orthologs of genes involved in mammalian itch transduction are also present in the zebrafish (*Kaslin and Panula, 2001*; *Pei et al., 2016*; *Xu et al., 2011*). Studying how these itch genes operate in the somatosensory system of zebrafish could reveal conserved itch transduction mechanisms, providing insight into the evolution of itch.

**eLife digest** Itch is a common and uncomfortable sensation that creates a strong desire to scratch. This mechanism may have evolved so animals can remove harmful parasites or substances from themselves. Feelings like touch, pain, and itch arise when stimuli such as mechanical pressure, temperature, or chemicals activate groups of specialized neurons in the skin. This response takes place when certain proteins – or receptors – at the surface of the neurons are stimulated. For instance, TRP ion channels such as TRPA1 play an important role in both the itch and pain responses. In mammals, directly activating these channels elicits pain. Itch is felt when itch responsive receptors are activated on a distinct set of neurons, which in turn activate TRP receptors. Although these processes have been well-studied in mammals, little is known about the existence of itch sensation in other animals.

To explore this, Esancy, Condon, Feng et al. exposed zebrafish to chemicals that induce itch in mammals, and found that imiquimod, a medicine used to treat certain skin conditions, can elicit itch in fish. When this chemical was injected into the lips of a fish, the animal rubbed them against the walls of its tank, akin to scratching an itch. Further experiments showed that imiquimod directly activated the pain-sensing ion channel TRPA1. In fact, this receptor was essential to the 'scratching' behavior: fish genetically engineered to lack TRPA1 did not react to the drug.

Fluorescent proteins were then used to track when the neurons that carry TRPA1 were activated. This revealed that, in the skin of zebrafish, there are at least two functionally distinct populations neurons that express TRPA1. One population, whose activation is associated with the animal 'scratching', responds even when TRPA1 receives a low level of stimulation. The other population is less sensitive: it responds only to high-intensity stimuli and is associated with a pain response such as freezing and slower movements. Further experiments in the mouse suggest that this mechanism is present in mammals as well. This coding strategy explains how pain and itch can be experienced when the same receptors are being activated.

Studying how animals like fish experience itch gives an insight into how detecting these sensations could have evolved. In turn, understanding this mechanism at the molecular and cellular levels may help find new ways to design better treatments for itch and pain disorders.
DOI: https://doi.org/10.7554/eLife.32036.002

## Results

### Imiquimod evokes itch in zebrafish

In an effort to determine if pruritic stimuli are capable of eliciting somatosensory activity in zebrafish, we screened five compounds known to both induce acute pruritus in mammals and act on receptors expressed by zebrafish (*Schön and Schön, 2007*; *Bell et al., 2004*; *Lieu et al., 2014*; *Yamaguchi et al., 1999*; *Tsujii et al., 2009*), excluding pruritogens that act on receptors that do not have a zebrafish ortholog, such as MRGPR agonists. Allyl isothiocyanate (AITC), a known algogen and TRPA1 agonist (*Prober et al., 2008*; *Jordt et al., 2004*), was used as the positive control. We evaluated somatosensory neuronal responses to pruritic compounds using transgenic larvae that pan-neuronally express the neuronal activity indicator CaMPARI (*elavl3*:CaMPARI), a fluorescent protein that permanently photoconverts from green to red in the presence of calcium and 405 nm light (*Fosque et al., 2015*). Using this approach, we were able to view trigeminal neuronal activity in 3 day post fertilization (dpf) larval zebrafish following the application of each pruritogen (*Figure 1A–H*). Of the pruritogens screened, only imiquimod (IMQ) significantly (p<0.05) activated zebrafish trigeminal ganglia (TG) neurons (*Figure 1I*).

We have previously reported that noxious stimuli evoke locomotion in larval zebrafish (*Gau et al., 2013*). When 5dpf larval zebrafish were exposed to individual pruritogens, only IMQ elevated baseline locomotion (p<0.001), producing a dose dependent increase (*Figure 1—figure supplement 1A*). When coupled with our findings in CaMPARI transgenics, these results indicate that IMQ likely acts through somatosensory neurons to evoke behavioral responses. While 5-HT did produce a reduction in locomotion, it did not activate somatosensory neurons (*Figure 1I,J*). Given the limited sensitivity of the locomotor assay we were unable to differentiate nocifensive behavior elicited by

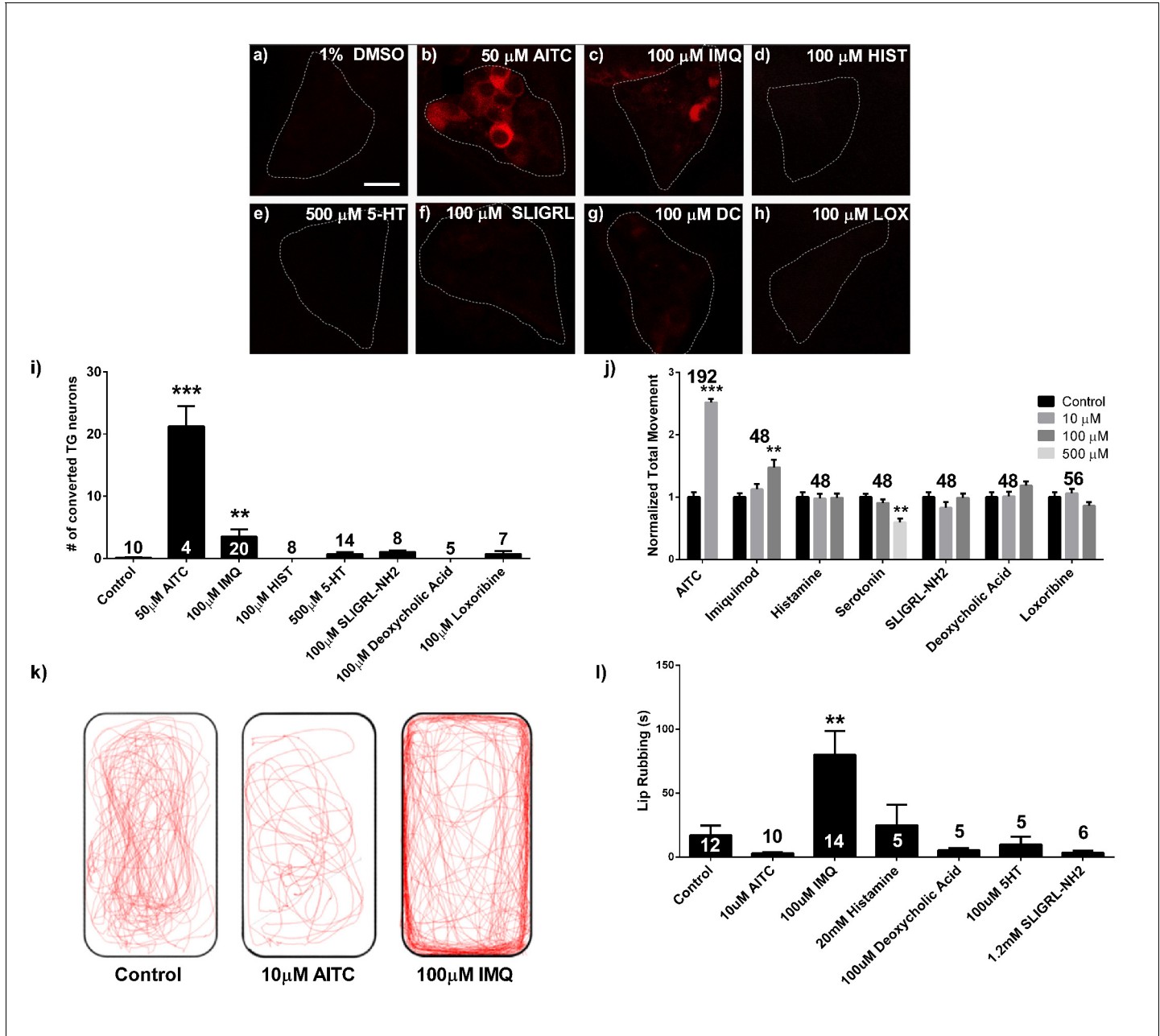

**Figure 1.** Imiquimod produces behavioral and neuronal responses in the zebrafish. (A–H) 3dpf live *elavl3*:CaMPARI TG imaging of larvae exposed to 405 nm light and control (A), 50 µM allyl isothiocyanate (AITC) (B), 100 µM imiquimod (IMQ) (C), 100 µM histamine (HIST) (D), 500 µM serotonin (5-HT) (E), 100 µM SLIGRL-NH2 (F), 100 µM deoxycholic acid (DCA) (G), and 100 µM loxoribine (LOX) solutions (H). (I), Counts of photoconverted cells from experiments (A-H). (J), 5dpf WT larval locomotor behavior screen. (K) Representative traces of adult behavior. (L) Adult lip-rubbing behavioral assay. (I, J, L) ***p<0.001, **p<0.01, one-way ANOVA. Bars represent mean ± s.e.m.

DOI: https://doi.org/10.7554/eLife.32036.003

The following figure supplement is available for figure 1:

**Figure supplement 1.** Behavioral effects of noxious pruritic and algogenic stimuli in the zebrafish.

DOI: https://doi.org/10.7554/eLife.32036.004

AITC from potentially pruritic behavior elicited by IMQ. To address this issue, we employed an adult zebrafish behavioral assay (*Correia et al., 2011*; *Maximino, 2011*). Injection of IMQ into the lip elicited a lip-rubbing behavior that may constitute a form of itch-scratch response in zebrafish, a behavior that was absent in sham-injected control fish (p<0.001) (*Figure 1K,L*; *Videos 1—3*) and distinct

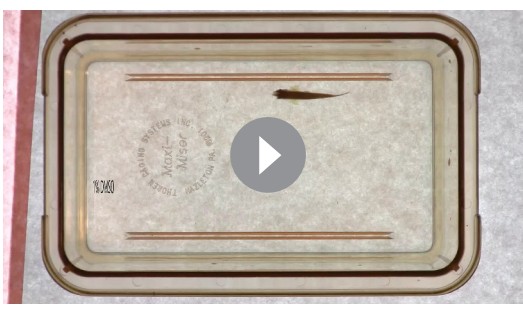

**Video 1.** Normal adult zebrafish swimming behavior. This video is an example of typical swimming behavior following a cutaneous injection of 1% DMSO into the upper lip of an adult zebrafish. Note that the zebrafish exhibits minimal interaction with the walls of the tank, and its swimming capacity is not compromised.
DOI: https://doi.org/10.7554/eLife.32036.005

from previously described zebrafish nocifensive, escape, or exploratory behaviors (*Colwill and Creton, 2011*; *Egan et al., 2009*; *Levin et al., 2007*). Consistent with studies of nocifensive behaviors, injection of AITC produced freezing behavior (p<0.01) (*Figure 1—figure supplement 1C*) as well as a significant decrease (p<0.05) in velocity not observed in control or IMQ injected fish (*Figure 1—figure supplement 1B*). Such distinct behavioral responses imply that zebrafish are capable of experiencing, and responding differentially to, discrete stimuli analogous to itch and pain in mammals.

## *Trpa1b*, but not *tlr7*, is required for both neuronal and behavioral responses to imiquimod

The mechanism by which IMQ elicits itch in mammals is unclear. IMQ, a treatment for various skin disorders, acts through TLR7 to stimulate an immune response, with intense itching and painful burning commonly reported as side effects (*Chang et al., 2005*; *Lebwohl et al., 2004*). In mice, however, IMQ is reported to be itch selective, and only elicits scratching, but not nociceptive, behaviors (*Liu et al., 2010*; *Kim et al., 2011*). Furthermore, there is dispute surrounding TLR7's role in IMQ-induced itch. One study found that $Tlr7^{-/-}$ mice showed deficits in IMQ evoked itch and proposed that *Tlr7* expressed in dorsal root ganglion (DRG) neurons was mediating IMQ transduction (*Liu et al., 2010*). TLR7 has also been reported to couple with TRPA1 in DRG neurons to evoke nociception in response to other TLR7 agonists (*Park et al., 2014*). However, conflicting studies found that $Tlr7^{-/-}$ mice exhibited no deficits in IMQ-evoked itch, and RNAseq analysis of DRG neurons found no evidence for *Tlr7* expression (*Kim et al., 2011*; *Usoskin et al., 2015*; *Li et al., 2016*). We therefore investigated whether TLR7 and/or TRPA1 were involved in transducing the neural and behavioral responses to IMQ in zebrafish.

*In situ* hybridization studies in zebrafish larvae revealed that *tlr7* mRNA expression is restricted to known hematopoietic regions in larval zebrafish (*Bresciani, 2014*; *Du et al., 2011*), and was notably absent in both TG and Rohon-Beard (RB) neurons (*Figure 2A*; *Figure 2—figure supplement 1I*). As

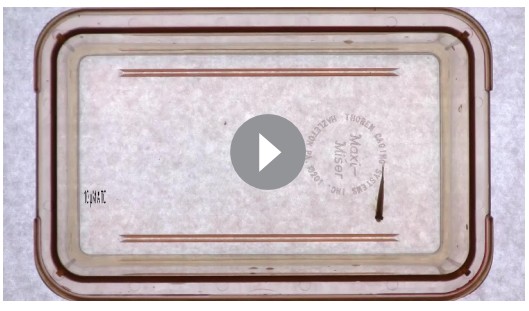

**Video 2.** Adult zebrafish injected with AITC demonstrate nocifensive behaviors. This video is an example of typical swimming behavior following a cutaneous injection of 10 μM AITC into the upper lip of an adult zebrafish. The zebrafish exhibits previously described nocifensive behaviors, such as dramatically reduced locomotion, bouts of freezing, and heightened respiration.
DOI: https://doi.org/10.7554/eLife.32036.006

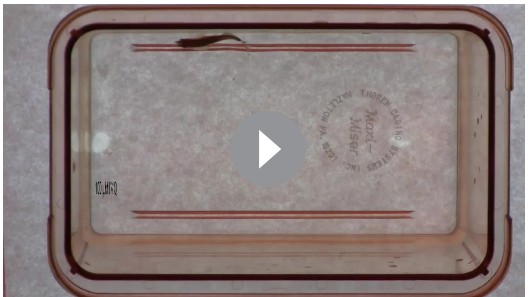

**Video 3.** Adult zebrafish injected with IMQ demonstrate pruritic behaviors. This video provides a typical example of swimming behavior observed in adult zebrafish following a cutaneous upper lip injection of 100 μM IMQ. The fish not only exhibits increased swimming velocity in response to this stimulus, but also frequently rubs its lips against the walls of its tank, engaging in a behavior that is potentially analogous to mammalian scratching.
DOI: https://doi.org/10.7554/eLife.32036.007

expected, *trpa1b* expression was observed in both TG and RB neurons (**Figure 2B**; **Figure 2—figure supplement 1C**). Therefore, any role TLR7 could play in IMQ-evoked behavior would be via indirect mechanisms.

To determine if *trpa1b* and/or *tlr7* are required for IMQ induced behaviors, we introduced early nonsense mutations in the coding sequences of both genes (**Kimura et al., 2014**). *Tlr7$^{-/-}$* zebrafish larvae exhibited a significant (p<0.001) increase in total locomotion when exposed to IMQ (100 µM) that was indistinguishable from controls (**Figure 2C**). However, *trpa1b$^{-/-}$* larvae demonstrated no response to IMQ (100 µM), while their WT siblings displayed normal IMQ induced behaviors (p<0.01, **Figure 2D**). These data support a mechanism where *trpa1b*, but not *tlr7*, is necessary for mediating behavioral responses to IMQ in larval zebrafish. As expected based on previous reports, behavioral responses to AITC were absent in *trpa1b$^{-/-}$* fish (**Figure 2—figure supplement 1F**) (**Prober et al., 2008**). Notably, *trpa1b$^{-/-}$* fish demonstrated an equivalent increase in locomotor behavior as their WT siblings when exposed to increased temperatures (**Figure 2—figure supplement 1E**). This indicates that the *trpa1b* mutation specifically affects Trpa1b-mediated sensations, rather than causing generalized sensory impairment.

To test whether the presence of Trpa1b was necessary to mediate neuronal responses in larval zebrafish TG, we conducted *in vivo* calcium imaging using *elavl3*:GCaMP5g larvae (**Akerboom et al., 2012**) (**Figure 2G**). In WT larvae, IMQ activation was seen exclusively in a subset of AITC responsive neurons (4/36, *n* = 5 larvae). *Trpa1b$^{-/-}$* fish, however, exhibited no response to either IMQ or AITC (0/89 total neurons, *n* = 5 larvae) (**Figure 2H**).

Notably, the highly specific TLR7 agonist loxoribine did not evoke TG neuron activation, larval locomotion, or adult lip-rubbing behavior, further strengthening the finding that Tlr7 does not play a role in IMQ evoked behaviors in zebrafish (**Figure 1I,J**; **Figure 1—figure supplement 1D**). This finding is similar to reports in the mouse demonstrating that loxoribine does not elicit pruritic behavioral responses (**Kim et al., 2011**).

## Imiquimod directly activates TRPA1

To determine if TRPA1 directly interacts with IMQ to produce itch, we utilized the TLR7-deficient cell line HEK293T (**Hornung et al., 2005**). HEK cells transfected with zebrafish, mouse, and human *Trpa1* showed a dose-dependent increase in intracellular calcium following application of IMQ and AITC that was not observed in HEK cells alone (**Figure 3A–F**; **Figure 3—figure supplement 1A,C**). Importantly, we found that loxoribine did not activate HEK cells transfected with zebrafish or mouse *Trpa1* (**Figure 3—figure supplement 2A–D**), indicating that TRPA1 is responsive to IMQ, and not to TLR7 agonists in general.

While we observed no expression of *tlr7* in the zebrafish TG (**Figure 2A**), given the lack of consensus over its functional role we sought to determine whether TLR7 might serve as a pruritic co-receptor that potentiates the TRPA1 response. We co-transfected HEK cells with *Trpa1* and *Tlr7* and examined the calcium responses following treatment with IMQ and observed no discernible differences in the average peak responses to IMQ between *Trpa1* and *Trpa1 + Tlr7* conditions (**Figure 3A–C**, **Figure 3—figure supplement 1C**). Whole cell electrophysiological experiments corroborated these findings. When stimulated with IMQ, voltage-clamped HEK cells transfected with mouse or zebrafish *Trpa1* demonstrated a significant increase in current (**Figure 3G**; **Figure 3—figure supplement 1G**). Co-transfecting mouse *Tlr7* with mouse *Trpa1* in HEK cells had no demonstrable effect on current influx (**Figure 3—figure supplement 1H**). Additionally, no difference was found in the IMQ current density dose-response curves for cells transfected with zebrafish *trpa1b*, mouse *Trpa1*, or mouse *Trpa1* + mouse *Tlr7* (**Figure 3H**). In contrast to previous reports, we found no evidence that TLR7 coupled with TRPA1 in the presence of loxoribine as measured by ratiometric calcium imaging and whole cell electrophysiology in mouse and zebrafish (**Figure 3—figure supplement 2A–D**) (**Liu et al., 2010**). Together, our data suggest that TLR7 plays no role in the direct activation of somatosensory neurons, and does not appear to potentiate the response of TRPA1 to IMQ.

Following these results, we confirmed that mouse and human TLR7 were present and functional in our assays (**Figure 3—figure supplement 1I–K**) (**Mitchell and Sugden, 1995**). Intriguingly, zebrafish Tlr7 did not respond to either IMQ or loxoribine in a dual-luciferase assay, suggesting that zebrafish Tlr7 is not activated by mammalian TLR7 agonists (**Figure 3—figure supplement 2E**). However, due to a lack of a Tlr7 positive control, we were unable to confirm that Tlr7 was functional in our heterologous expression system. With this caveat in mind, the lack of zebrafish Tlr7 response to

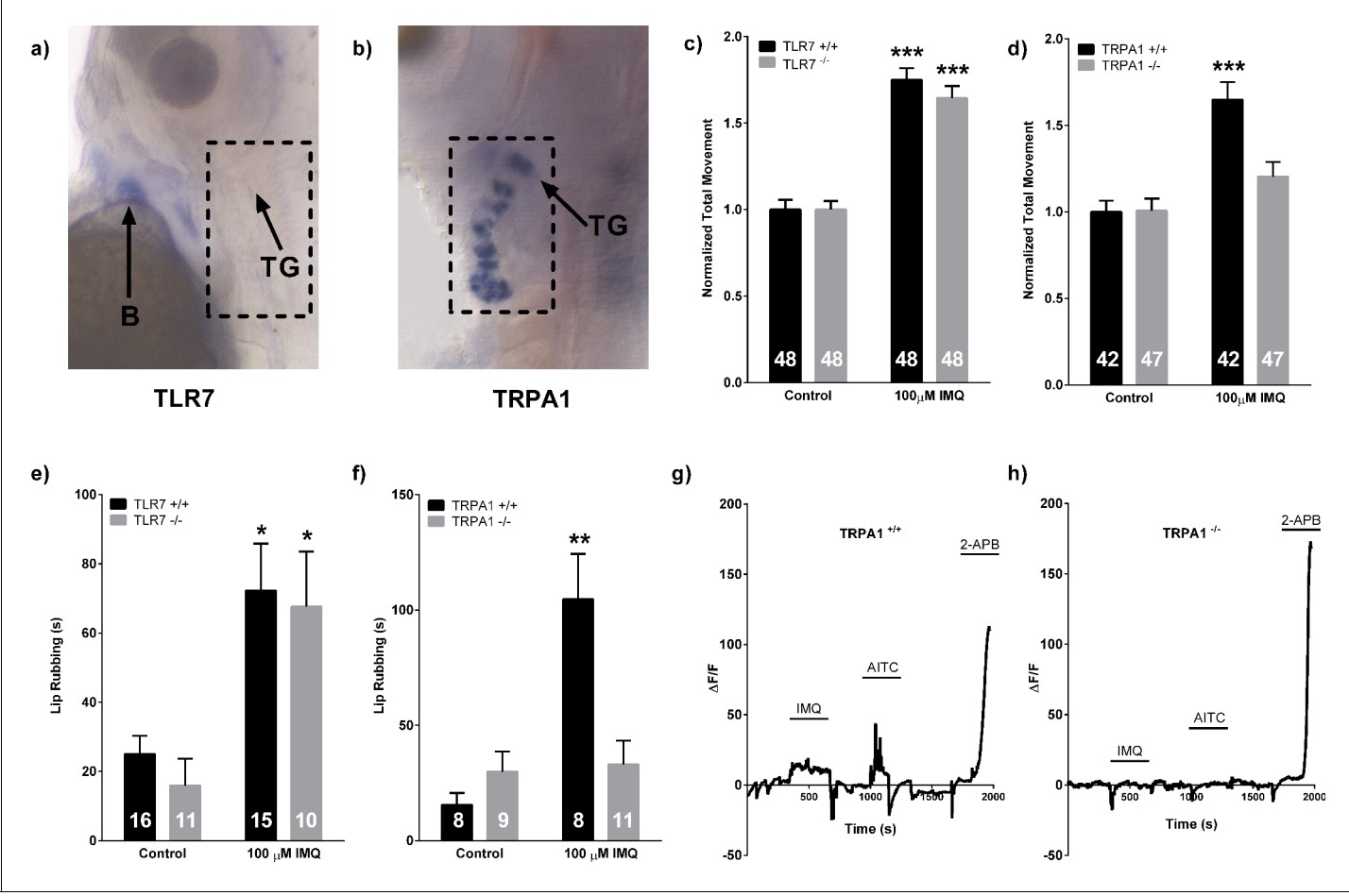

**Figure 2.** Trpa1b, but not Tlr7, is required for both neuronal and behavioral responses to IMQ. (A, B) *In situ* hybridization in 3dpf WT larvae probing for *tlr7* and *trpa1b* mRNA respectively. (C, D) Larval locomotor assay of 5dpf *tlr7^{+/+}* and *tlr7^{-/-}* (C) or *trpa1b^{+/+}* and *trpa1b^{-/-}* (D) larvae. (E, F) Adult lip-rubbing behavioral assay of *tlr7^{+/+}/tlr7^{-/-}* (E) and *trpa1b^{+/+}/trpa1b^{-/-}* (F) fish. (C), (D), (E), (F), 100 µM IMQ used. ***p<0.001, **p<0.01, Student's *t*-test. Bars represent mean ± s.e.m. (G, H) Representative calcium imaging traces of 3dpf *trpa1b^{+/+}* (G) and *trpa1b^{-/-}* (H) larvae in a transgenic *elavl3*HuC: GCaMP5 background exposed to 100 µM IMQ, 50 µM AITC, and 1 mM 2-APB. B = blood, TG = trigeminal ganglion.

DOI: https://doi.org/10.7554/eLife.32036.008

The following figure supplement is available for figure 2:

**Figure supplement 1.** TRPA1b and TLR7 nonsense mutations in the zebrafish.

DOI: https://doi.org/10.7554/eLife.32036.009

these TLR7 agonists lends further credence to the conclusion that Tlr7 is not involved in somatosensory neuronal activation or behavior in this species.

If TRPA1 does not couple with TLR7, but is instead directly activated by both IMQ and AITC, how could IMQ be itch-selective in the mouse? To address this question we assessed the $EC_{50}$ and peak responses to IMQ and AITC in HEK cells transfected with *Trpa1* from different species. The $EC_{50}$ of IMQ for zebrafish, mouse and human TRPA1 was consistently higher than the AITC $EC_{50}$, demonstrating that IMQ is a weaker agonist than AITC (**Figure 3—figure supplement 1K**). Notably, while the IMQ $EC_{50}$ of zebrafish Trpa1 and human TRPA1 were only 2–3 fold greater than that of the $EC_{50}$ of AITC, mouse TRPA1 demonstrated a ~40 fold difference between the $EC_{50}$ of AITC and IMQ. Furthermore, only mouse TRPA1 elicited significantly lower maximum responses to IMQ than AITC (**Figure 3D–F**). In similar electrophysiology experiments, HEK cells transfected with zebrafish *trpa1b* exhibited identical current density responses upon stimulation with the maximum dose of either AITC or IMQ, but cells transfected with mouse *Trpa1* exhibited significantly higher current density responses following stimulation with AITC, relative to IMQ (**Figure 3I**). Such physiological differences

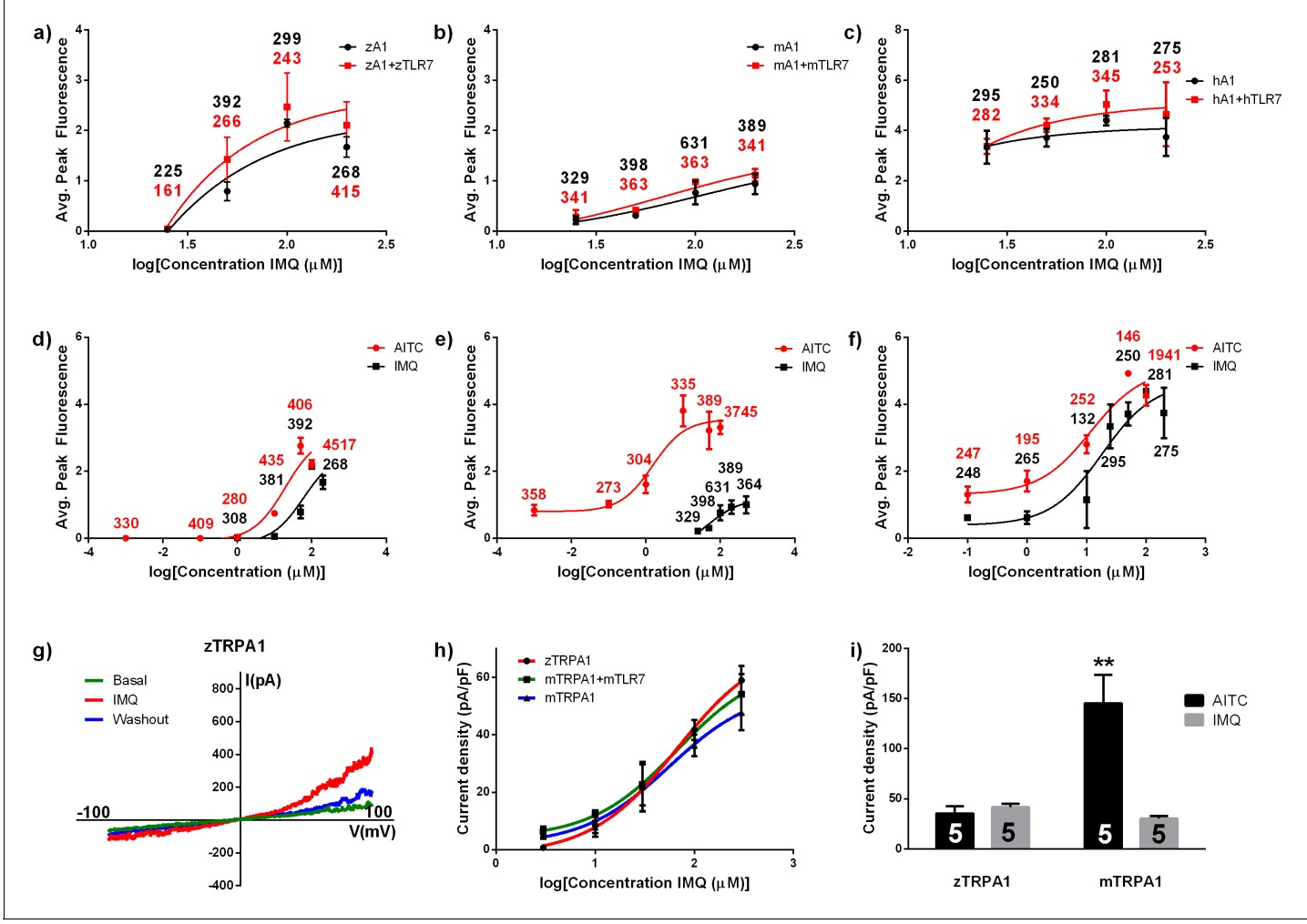

**Figure 3.** Imiquimod directly activates TRPA1. (A–C) Calcium imaging of HEK cells transfected with zebrafish (A), mouse (B), and human (C) *Trpa1* or *Trpa1 + Tlr7*. (D, E, F) Calcium imaging dose response curves of HEK cells transfected with zebrafish (D), mouse (E), and human (F) *Trpa1*, exposed to IMQ or AITC. (A–F) Numbers represent total cell counts per condition. (G) Patch clamp of HEK cell transfected with zebrafish *trpa1b* exposed to 100 μM IMQ. (H) Patch clamp dose response curve for HEK cells transfected with zebrafish *trpa1b*, mouse *Trpa1*, or mouse *Trpa1 + mouse Tlr7*. (I) Current density values of HEK cells exposed transfected with zebrafish *trpa1b* or mouse *Trpa1* and exposed to 100 μM AITC or 100 μM IMQ. (G–I) n = 5 cells per condition. **p<0.01, Student's *t*-test. Bars represent mean ± s.e.m.

DOI: https://doi.org/10.7554/eLife.32036.010

The following figure supplements are available for figure 3:

**Figure supplement 1.** Transfected HEK cell gene expression and functionality.
DOI: https://doi.org/10.7554/eLife.32036.011

**Figure supplement 2.** Loxoribine effectively stimulates *Tlr7*-transfected HEK cells but does not evoke intracellular calcium flux.
DOI: https://doi.org/10.7554/eLife.32036.012

in TRPA1 function between species could provide a potential mechanism for the itch selectivity of IMQ in mice, implying that IMQ is not a strong enough mouse TRPA1 agonist to elicit nociception in this species. Conversely, the ability of both zebrafish and human TRPA1 to respond equally to AITC and IMQ at maximal doses potentially explains how IMQ can elicit both itch and pain sensations in humans, and suggests that the same may be observable in fish.

## Imiquimod responsive neurons are a primed subpopulation of TRPA1-expressing neurons

In the preceding *in vivo* calcium imaging experiments, we observed that only a small proportion of zebrafish Trpa1+ neurons, identified by their responsiveness to AITC, were also responsive to the

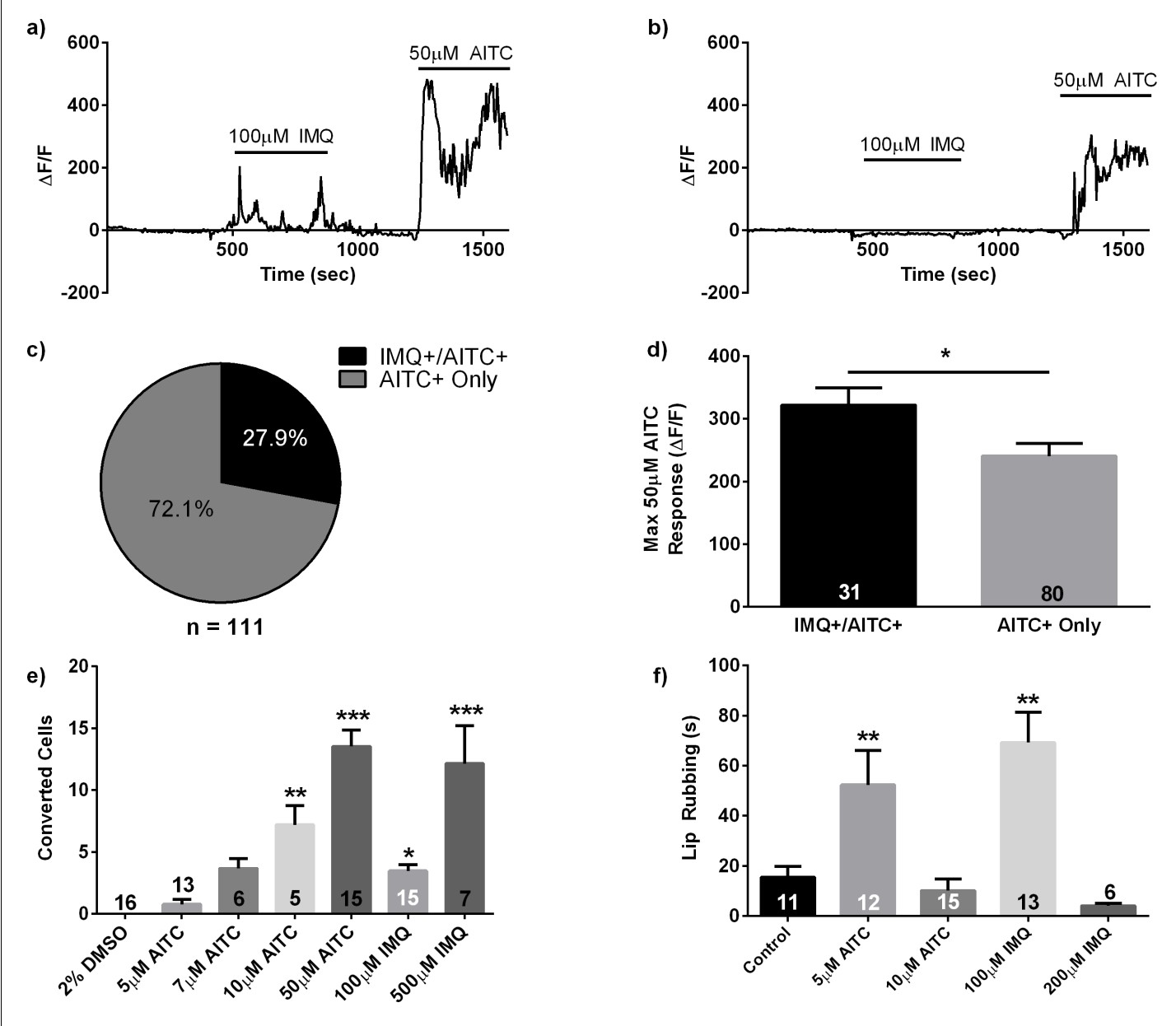

**Figure 4.** Imiquimod responsive cells are a primed subpopulation of TRPA1 positive cells and stimulus intensity affects behavioral and neuronal responses. (A, B) Representative traces from calcium imaging experiments in 3dpf *elavl3*:H2BGCaMP6 zebrafish exposed to 100 μM IMQ and 50 μM AITC. An IMQ+/AITC+ TG neuron is shown in (A), while an AITC+ only neuron is shown in (B). (C) Quantification of neuronal subtypes within AITC + neurons. *n* = 31 IMQ+/AITC+ neurons, *n* = 80 AITC+ only neurons. (D) Comparison of the maximum ΔF/F during the 50 μM AITC stimulus between neuronal subtypes. (E) Number of photoconverted neurons in 3dpf *elavl3*:CaMPARI zebrafish TG following stimulation with TRPA1 agonists. (F) Quantification of lip-rubbing behavior in adult zebrafish following exposure to TRPA1 agonists. Bars represent means ± s.e.m. *p<0.05, **p<0.01, ***p<0.001, Student's *t*-test (D), one-way ANOVA (E, F).

DOI: https://doi.org/10.7554/eLife.32036.013

The following figure supplements are available for figure 4:

**Figure supplement 1.** Further effects of stimulus intensity on zebrafish neuronal activity and behavior.
DOI: https://doi.org/10.7554/eLife.32036.014

**Figure supplement 2.** Imiquimod activates a specific subset of neurons in a non-stochastic manner in the zebrafish.
DOI: https://doi.org/10.7554/eLife.32036.015

IMQ stimulus. To further explore this result, we used *elval3*:H2BGCaMP6 (*Chen et al., 2013*) transgenic zebrafish to record the response properties of larval TG neurons to IMQ and AITC. No IMQ+/AITC- neurons were found across 13 larvae. Among AITC+ neurons, 28% (31/111) were responsive to IMQ (*Figure 4A–C*).

The above data affirms that IMQ+ neurons are a subset within a larger population of Trpa1+ TG neurons, implying that a population coding strategy for pruritus might be at play. This does not itself answer the question of how such an itch-selective Trpa1+ subpopulation might be activated by a TRPA1 agonist without recruiting other Trpa1+ neurons that may code for nociceptive behaviors. Based on our finding that IMQ is a weaker TRPA1 agonist than AITC, one potential explanation is that such an itch-selective Trpa1+ population is more sensitive to TRPA1 agonists, and can be activated by weaker (pruritic) stimuli. In IMQ+/AITC+ neurons, we determined that the average maximum fluorescence intensity response to the IMQ stimulus was significantly lower than that of the AITC stimulus (p<0.001), implying that at the concentrations used, IMQ is indeed a weaker TRPA1 stimulus than AITC in vivo (*Figure 3—figure supplement 2A*). Furthermore, we found that the maximum AITC response of IMQ+/AITC+ neurons was significantly greater than that of AITC+ only neurons (*Figure 4D*). Likewise, IMQ+/AITC+ neurons displayed a significantly greater average AITC response than AITC+ only neurons (p<0.05, *Figure 4—figure supplement 1B*).

These data suggest that IMQ+ TG neurons are primed to respond to TRPA1 agonists and support a model where relatively weak TRPA1 stimuli, such as IMQ at the concentration used, could selectively recruit a potential itch-coding subpopulation of Trpa1+ neurons. Higher intensity stimuli like AITC at the concentrations used, however, would activate the majority of Trpa1+ neurons to evoke nocifensive behaviors, positively correlating with findings that nociception takes precedence over itch sensation (*Roberson et al., 2013*; *Ross et al., 2010*; *Liu et al., 2011*).

To verify that IMQ activates a selective subset of Trpa1+ neurons as opposed to activating Trpa1 + neurons stochastically, we performed calcium imaging experiments in which 3dpf *elavl3*:H2BGCaMP6 were exposed to successive pulses of 100 µM IMQ. Of the IMQ+ neurons we identified across five fish, 92.3% (12/13) responded to both pulses of IMQ, whereas only 7.7% (1/13) responded only to the second pulse of IMQ (*Figure 4—figure supplement 2A–C*). Additionally, it is possible the single neuron that responded only to the second pulse of IMQ may also be a dual-responder. While the GCaMP fluorescence change only crossed our response threshold during the second pulse, it is possible that the sloping baseline may have obscured a minimal response to the first pulse, especially considering the low amplitude of the second response. However, for purposes of completion we decided to include this trace in our final counts. Given the finding that only ~30% of AITC responsive neurons responded to one pulse of IMQ (100 µM), if one assumes that this is the probability that any given AITC responsive neuron would respond to IMQ (100 µM), and that responses to IMQ are stochastic in nature, one would expect only a small fraction of neurons to be double responders (~9%) (*Figure 4D*). The finding that nearly all IMQ-responsive neurons were dual responders argues that these neurons comprise a distinct population of Trpa1 expressing neurons, primed to respond to low intensity TRPA1 dependent stimuli.

## Stimulus intensity affects behavioral and neuronal responses

Although the IMQ dose-response curve in zebrafish *trpa1b*-transfected HEK cells was rightward shifted, it was eventually able to elicit the same amount of intracellular calcium flux that AITC evoked. This implies that at a sufficiently high concentration, IMQ might be able to recruit neurons outside of the IMQ+ subpopulation observed in the above larval calcium imaging experiments, thus eliciting neuronal and behavioral responses characteristic of AITC at nociceptive concentrations. Likewise, it is also possible that at low enough concentrations, AITC is capable of eliciting the pruritic neuronal and behavioral responses we observed following application of IMQ.

In CaMPARI fish we observed that decreasing the concentration of applied AITC correlated with a reduction in the number of photoconverted neurons (*Figure 4E*). Additionally, administering higher IMQ concentrations converts an equivalent number of neurons as high concentrations of AITC (*Figure 4E*), suggesting that for TRPA1 agonists, eliciting pruritus or nociception is dependent more on stimulus intensity than identity.

*In vivo* GCaMP imaging bolstered our CaMPARI findings that stimulus intensity affects which subpopulations of Trpa1+ neurons are activated. In these experiments, we observed that increasing the stimulus intensity activated more neurons. Only a subset of neurons that responded to a high

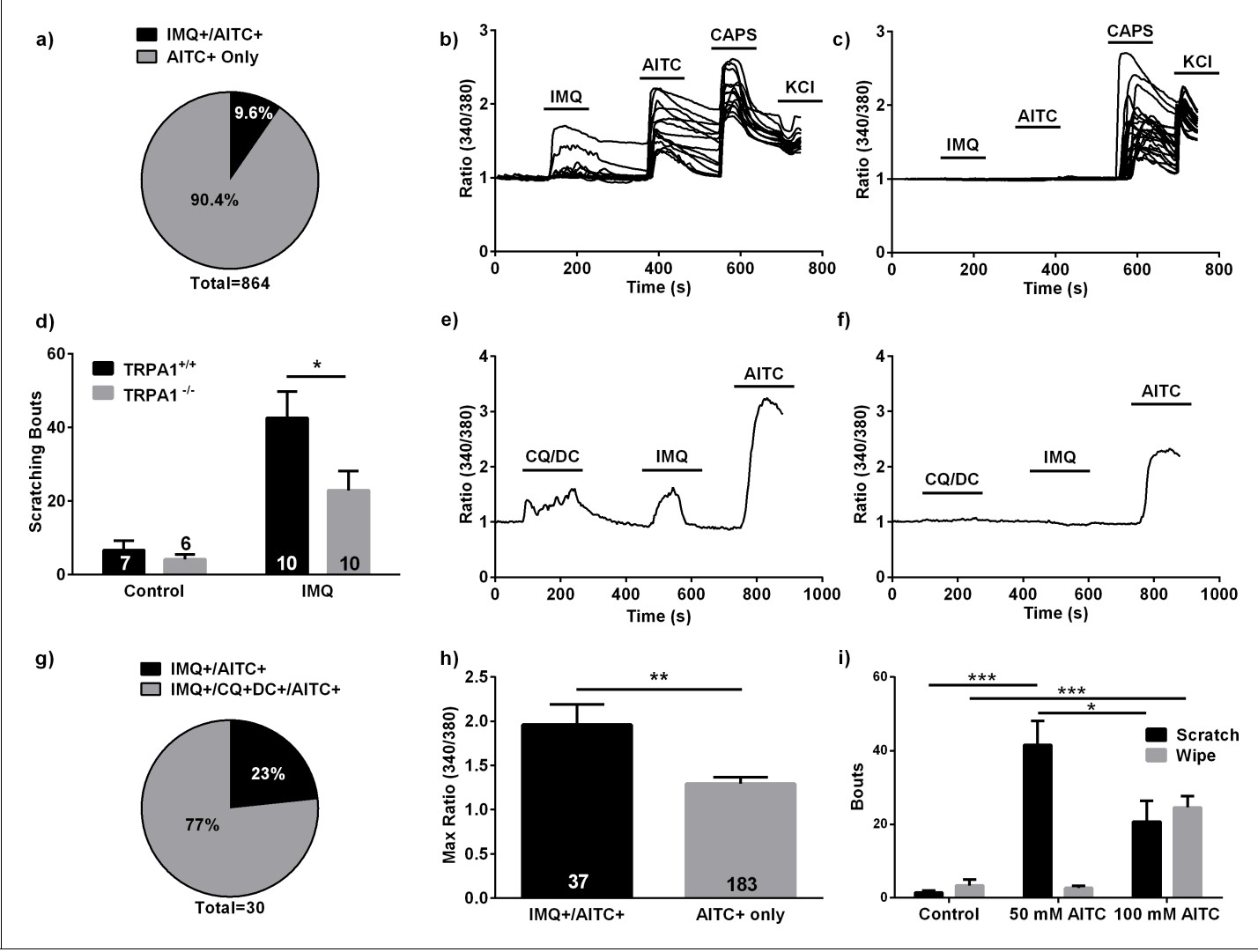

**Figure 5.** Imiquimod's effects in the mouse. (A) Proportions of IMQ+/AITC+ and AITC+ only DRG neurons observed in calcium imaging experiments (n = 83 IMQ+/AITC+ neurons, n = 781 AITC+ only DRG neurons). (B, C) Representative traces from calcium imaging experiments on dissociated mouse DRG neurons from WT (B) (n = 15 neurons) and $Trpa1^{-/-}$ (C) (n = 23 neurons) mice. (A–C), 100 μM IMQ and 100 μM AITC used. (D) Quantification of scratching bouts in WT and $Trpa1^{-/-}$ mice following IMQ (10 μg) injections. (E, F) Representative traces from calcium imaging experiments in which 100 μM IMQ, 100 μM CQ, 100 μM DC, and 100 μM AITC were applied. IMQ+/CQ+DCA+/AITC+ (e) and AITC+ only (f) neurons are shown. (G) Quantification of the IMQ+ neuronal populations from the experiments shown in (E, F) (n = 27 IMQ+/CQ+DCA+/AITC+ neurons, 7 IMQ+/AITC + neurons). (H) Comparison of maximum ΔF/F between IMQ+/AITC+ and AITC+ only. 100 μM IMQ, 10 μM AITC used. All calcium imaging experiments performed on dissociated mouse DRG neurons. (I) Quantification of scratching bouts in WT mice following injections of varying AITC concentrations. *p<0.05, **p<0.01, ***p<0.001, Student's *t*-test. Bars are expressed as mean + s.e.m.

DOI: https://doi.org/10.7554/eLife.32036.016

The following figure supplement is available for figure 5:

**Figure supplement 1.** Calcium imaging of discrete neuronal subpopulations in the mouse.

DOI: https://doi.org/10.7554/eLife.32036.017

concentration (50 μM) of AITC responded to a lower concentration (10 μM) of AITC (11/42), and of those even fewer neurons responded to 100 μM IMQ (4/11) (*Figure 4—figure supplement 1C*). Furthermore, increasing the IMQ concentration to 200 μM in adult behavioral experiments evoked nocifensive behaviors such as elevated freezing and significantly reduced velocity, and the itch-like lip rubbing behavior seen at 100 μM was notably absent (*Maximino, 2011*; *Sneddon, 2009*) (*Figure 4F*; *Figure 1—figure supplement 1C*, *Figure 4—figure supplement 1D*). Conversely, low concentrations of AITC (5 μM) elicited both itch-like lip rubbing behavior and increased velocity

(*Figure 4F*; *Figure 4—figure supplement 1C*). These data indicate that the subpopulation of Trpa1 + neurons that drive itch behavior in the zebrafish are distinct in their sensitivity to TRPA1 agonists, but can be activated by either AITC or IMQ at the appropriate concentration to produce equivalent behaviors.

## TRPA1 mediates itch behavior and neuronal responses in in the mouse

We found that IMQ elicited responses in 9.6% (83/864) of cultured AITC+ DRG neurons (*Figure 5A*) from WT animals. To determine if TRPA1 mediates IMQ responses in mice, we examined IMQ-evoked responses in DRG neurons from both WT and *Trpa1$^{-/-}$* animals using ratiometric calcium imaging. We found that both IMQ and AITC responses were completely abolished in DRG neurons obtained from *Trpa1$^{-/-}$* animals, while neurons from WT siblings exhibited normal responsivity to both stimuli (*Figure 5B,C*). Furthermore, application of loxoribine did not elicit calcium responses in mouse DRG neurons, providing evidence that TLR7 stimulation does not result in activation of somatosensory neurons (*Figure 5—figure supplement 1H*). We next explored whether IMQ-evoked scratching behavior was also dependent on TRPA1. High-dose IMQ (125 µg) paw injections did not evoke nocifensive behaviors in WT mice (n = 0/10 IMQ-injected), consistent with previous reports (*Kim et al., 2011*). However, scratching bouts at a low concentration of IMQ (10 µg, nape injected) were significantly attenuated in *Trpa1$^{-/-}$* mice, demonstrating that TRPA1 is required for normal IMQ-induced scratching behavior (*Figure 5D*). Interestingly, a higher dose of IMQ (50 µg) evoked equivalent scratching behavior in both WT and *Trpa1$^{-/-}$* mice (*Figure 5—figure supplement 1G*). This result, taken together with our finding that isolated mouse DRG neuron responses to IMQ are TRPA1-dependent, suggests that IMQ can also evoke itch via indirect activation of somatosensory neurons, perhaps downstream of an immune response (*Bautista et al., 2014*; *Hoon, 2015*).

Given the itch selectivity of IMQ in the mouse, we sought to determine whether IMQ+ neurons were part of a population of DRG neurons that encode TRPA1-dependent pruritus. We therefore measured the overlap of IMQ+ neurons with DRG neurons that responded to a mixture of the TRPA1-dependent pruritogens deoxycholic acid (DC) and chloroquine (CQ) (*Figure 5E–F*) (*Tsujii et al., 2009*; *Wilson et al., 2011*; *Liu et al., 2009*). We found that the vast majority of IMQ + neurons (73%, 23/30) also responded to these pruritic stimuli (*Figure 5G*), indicating that in the mouse, IMQ+ neurons belong to a subpopulation of itch-encoding neurons.

Due to the parallels noted between zebrafish and mouse IMQ responses, we proceeded to investigate whether the correlation between stimulus intensity and neuronal activation that we observed in the zebrafish was conserved in the mouse. In the mouse, increasing the concentration of AITC activated more DRG neurons in a dose-dependent manner (*Figure 5—figure supplement 1I*). Furthermore, within the population of neurons that responded to lower concentrations of AITC, the IMQ + subpopulation was enriched (*Figure 5—figure supplement 1J*). We also found that IMQ+ neurons in the mouse had a smaller peak responses to IMQ than AITC (*Figure 5—figure supplement 1A– C*). Additionally, AITC peak responses within the IMQ+ population were significantly greater than in the AITC+ only population (p<0.01), demonstrating that the heightened sensitivity of IMQ+ neurons to TRPA1 agonists is conserved in the mouse (*Figure 5H*). Subsequent experiments revealed that IMQ+ DRG neurons exhibited a significantly higher maximum response to the TRPV1 agonist capsaicin (CAPS) than CAPS+ only neurons (*Figure 5—figure supplement 1D–F*), implying that the IMQ + neurons may be intrinsically more sensitive to noxious stimuli, not exclusively TRPA1 agonists. Finally, in accordance with our zebrafish data, we observed that AITC stimulus intensity dictates whether mice exhibit pruritic or nocifensive behaviors. In order to discriminate between nocifensive and pruritic behaviors, we employed a 'cheek model of itch' assay in which compounds injected into the cheek may elicit scratching (a pruritic response) or wiping (a nocifensive response) (*Shimada and LaMotte, 2008*; *Akiyama et al., 2010*). We observed that AITC (50 mM) produces significant scratching behavior with no appreciable wiping behavior (p<0.001 and N.S. respectively), while a higher dose of AITC (100 mM) results in a significant attenuation of the observed scratching behavior, as well as a significant wiping behavior (p<0.05 and p<0.001 respectively) (*Figure 5I*).

## Discussion

The sense of itch in mammals has been thoroughly explored, but there is a dearth of knowledge on the presence of itch, and the neural mechanisms which transduce it, in lower vertebrates such as

fish. Here, we use a popular model organism, the zebrafish, to demonstrate that fish are potentially capable of experiencing a form of rudimentary itch in response to the pruritogen IMQ, and suggest that this sensation is conveyed by the direct activation of TRPA1 on a specialized subset of somatosensory neurons highly sensitive to TRPA1 agonists.

Through a combination of experimental procedures, we show that the pruritogen IMQ can elicit behavior and neuronal responses in zebrafish. In larvae, IMQ elicits an increase in locomotion and activation of somatosensory TG neurons, suggesting that the observed locomotor effects likely originate in the periphery, and are not due to the direct action of IMQ upon more central structures. Furthermore, in adult zebrafish, peripherally applied IMQ produces a scratching-analogous lip-rubbing behavioral response that is distinct from the nocifensive behaviors evoked by the algogen AITC as well as previously described escape behaviors (*Correia et al., 2011*; *Maximino, 2011*; *Colwill and Creton, 2011*; *Egan et al., 2009*; *Levin et al., 2007*). The fact that IMQ-evoked behavioral responses manifest as lip-rubbing, as opposed to nocifensive or escape behaviors, further suggests that these fish are indeed experiencing an itch-like sensation. For animals whose anatomy prohibits scratching behavior with claws, the use of another body part or an external object is perhaps the only recourse to address a pruritic stimulus. Such scratching behaviors have previously been documented in animals from a variety of taxa (*Chevalier-Skolnikoff and Liska, 1993*; *Huber et al., 2008*; *Deecke, 2012*; *Green and Mattson, 2003*; *Delius, 1988*).

There is currently a lack of consensus in the literature as to how IMQ induces itch in mammals, although all studies show that TRPV1-expressing neurons are required (*Liu et al., 2010*; *Kim et al., 2011*). A major point of contention surrounds the involvement of the immune receptor TLR7, the therapeutic target of IMQ. Our work provides additional evidence that TLR7 is not involved in the direct activation of somatosensory neurons by IMQ (*Kim et al., 2011*; *Usoskin et al., 2015*; *Li et al., 2016*). Instead, the TRPA1 ion channel appears to play a key role in transducing IMQ-evoked neuronal effects and the behaviors that result from such neuronal activity. Unlike *trpa1b*, *tlr7* is not expressed in zebrafish somatosensory tissue, as demonstrated by *in situ* hybridization. Moreover, zebrafish that lacked functional TLR7 behaved equivalently to wildtype animals in response to IMQ application. By contrast, behavioral and neuronal responses to IMQ in both fish and mice were largely dependent upon TRPA1. Utilizing both electrophysiology and calcium imaging, we showed that IMQ was able to directly activate TRPA1, and that co-transfection of *Tlr7* did not potentiate IMQ-evoked responses. However, greater concentrations of IMQ were required to elicit the same levels of calcium flux that were evoked by lower concentrations of AITC, implying that IMQ is a weaker agonist than AITC in all species we examined. With zebrafish and human TRPA1, IMQ eventually reached the same maximal calcium response levels as AITC, while IMQ responses with mouse TRPA1 plateaued at lower maximal levels than AITC. This suggests that in the mouse, IMQ is incapable of evoking the same intensity of neuronal responses as AITC can, even at high concentrations. Our findings may provide a molecular explanation for previously-documented observations that in the mouse, IMQ appears to be itch-selective, while in humans, IMQ has been reported to elicit sensations of both itch and pain (*Chang et al., 2005*; *Lebwohl et al., 2004*; *Liu et al., 2010*; *Kim et al., 2011*). Findings from these *in vitro* experiments may also underlie our own observations in both mice and adult zebrafish behavioral experiments. In the context of TRPA1 activation, stronger stimuli elicited nocifensive behaviors, whereas weaker stimuli evoked pruritic behaviors. Furthermore, the effects of stimulus intensity upon neuronal activation and behavior were largely but not entirely independent of stimulus identity. In adult zebrafish low concentrations of IMQ evoked pruritic behaviors while higher concentrations of IMQ were capable of eliciting nocifensive freezing behaviors. In line with previous reports, we were unable to elicit murine nocifensive behaviors with high concentrations of IMQ, which we hypothesize was due to the inability of IMQ to maximally activate mouse TRPA1 (*Liu et al., 2010*; *Kim et al., 2011*). In contrast, AITC evoked behaviors were not species dependent as predicted by the dose response curves of zebrafish Trpa1 and mouse TRPA1 to this agonist. Low concentrations of AITC produced scratching behaviors, while higher concentrations of AITC evoked nocifensive behaviors in both species. While the direct activation of a TRP channel to evoke itch by a pruritogen or algogen is somewhat surprising, as it deviates from canonical pathways requiring both pruritic GPCRs and TRP ion channels, such a mechanism supports observations of itch in both human and mouse studies following application of typically noxious TRP channel agonists (*Akiyama et al., 2010*; *Højland et al., 2015*; *Sikand et al., 2009*).

Intriguingly, our observation that behaviors transitioned from a pruritic to a nocifensive pheno-type as stimulus intensity increased was mirrored by our observations in calcium imaging studies. Experiments with two different calcium indicators of neuronal activity, GCaMP and CaMPARI, revealed a correlation between TRPA1 agonist stimulus intensity and neuronal recruitment. Specifi-cally, stronger stimuli (high concentrations of TRPA1 agonists) activated more neurons than weaker stimuli (lower concentrations of TRPA1 agonists).

Based on these findings we hypothesized that a population coding strategy was being employed. In our model, low intensity stimuli would selectively activate a subset of TRPA1-expressing neurons that encode pruriception, whereas higher intensity stimuli would activate another population of TRPA1-expressing neurons that encode nociception. This hypothesis would be in line with recent findings that pruriception is encoded by different populations of somatosensory neurons than those that encode nociception (*Hoon, 2015*; *Roberson et al., 2013*; *Goswami et al., 2014*). Calcium imaging of zebrafish and mouse somatosensory neurons revealed the existence of at least two func-tionally distinct neuronal subpopulations. In both species, we observed a population of neurons that responded to both low and high intensity TRPA1 agonists and another that only responded to high intensity TRPA1 agonists. Our observation that sequential pulses of IMQ (100 µM) activate the low intensity TRPA1 agonist responsive population in zebrafish demonstrates that this population is likely genetically defined, and not determined stochastically. Furthermore, in the mouse, we observed that the vast majority of IMQ (100 µM) responsive neurons also responded to the TRPA1-dependent pru-ritogens chloroquine and deoxycholic acid, supporting the conclusion that these neurons constitute a pruritic subpopulation. Unfortunately, equivalent experiments could not be performed in the zebrafish, as we were unable to identify other pruritogens for this species.

In alignment with these results, in both species, neurons that responded to low intensity TRPA1 agonists also exhibited a greater response to high intensity AITC stimulation than neurons that only responded to high intensity AITC stimulation. This implies that putative itch-encoding neurons com-prise a more sensitive subset of somatosensory neurons than nociceptive neurons. Additionally, in both species, IMQ (100 µM) responsive neurons were enriched within populations of neurons that responded to low concentrations of AITC; in other words, we observed a high degree of overlap in the populations of neurons that responded to low intensity TRPA1 agonists irrespective of stimulus identity. Together, this provides further evidence of the greater sensitivity of a population of itch-encoding neurons to TRPA1 agonists.

Itch is a complex physiological phenomenon, and several models have been proposed to explain its transduction in the periphery and beyond (*Bautista et al., 2014*; *Hoon, 2015*; *Ross, 2011*; *Ma, 2010*; *Schmelz, 2010*; *McMahon and Koltzenburg, 1992*). Here, we describe a mechanism through which acute IMQ-evoked itch is mediated by the peripheral somatosensory neurons of two species. Weak stimuli such as lower concentrations of the TRPA1 agonists IMQ and AITC activate only a subset of highly sensitive pruriceptive TRPA1-expressing neurons, while the remaining nociceptive TRPA1-expressing neurons are only recruited by higher intensity TRPA1 stimuli. Our pro-posed model clearly is not the sole mechanism by which itch is coded as a distinct sensation from pain (*Bautista et al., 2014*; *Hoon, 2015*; *Ross, 2011*). However, in the context of pruritogens and algogens that are direct TPRA1 agonists, our findings overwhelmingly support a population-coding model for the discrimination of itch and pain.

Our data implicates a mechanism for the direct activation of TRPA1 by IMQ to elicit sensations of itch in both zebrafish and mice. Our findings that IMQ-induced itch was attenuated, but not abol-ished, in *Trpa1*[-/-] mice supports additional mechanisms for IMQ-evoked itch in this species. One may speculate that the activation of an immune response by IMQ in other cell types could result in a cas-cade of downstream signaling that could ultimately lead to a release of endogenous pruritic stimuli (e.g. histamine and serotonin from mast cells). In the zebrafish, TLR7 is unlikely to contribute even indirectly to IMQ-induced itch sensation, as we observed no behavioral phenotype in *tlr7*[-/-] animals. However, this does not rule out the possibility that pathogen exposure could stimulate the immune system via TLR activation or other pathways, leading to an indirect activation of pruritic somatosen-sory neurons and triggering both itch sensations and scratching behaviors. Several studies have implicated that Tlrs in several species of fish play a role in immunity (*Meijer et al., 2004*; *Kanwal et al., 2014*; *Zhou and Sun, 2015*; *Tanekhy et al., 2010*; *Kileng et al., 2008*; *Yang et al., 2012*). However, no study has explicitly queried potential connections between such immune responses and downstream neuronal activation or behavior in any fish species. While our results

demonstrate that TLR7 is not involved in transducing IMQ-evoked itch in the zebrafish, investigating the role of immune responses in mediating somatosensations (both in zebrafish and other species) would be an interesting route for future exploration.

It is plausible that an unknown receptor is activated by IMQ and couples with TRPA1 *in vivo* to evoke a somatosensory neuron response. Our TRPA1 mutant experiments would be unable to detect the existence of such a receptor, which would presumably not function to elicit neuronal activity in the absence of TRPA1. However, our *in vitro* experiments suggest that the existence of such an unknown receptor is unlikely, as it would also have to be endogenously expressed in HEK cells. Furthermore, the dose-dependent overlap we observed in the number of neurons that were responsive to IMQ and AITC, as well as similarities in the dose dependent itch and nociceptive behaviors elicited by these compounds in both zebrafish and mice, implies that these compounds are acting via the direct activation of TRPA1.

We observed that out of several mammalian pruritogens, only IMQ elicited behavioral and neuronal responses in zebrafish. While we tested pruritogens that targeted mammalian pruritic receptors with a known zebrafish ortholog, it is possible that other stimuli we did not test may also evoke sensations of itch in these animals. It is likewise possible that histamine, serotonin, TGR5, and PAR-2 receptors do not function in the itch pathway as pruritic receptors on primary sensory neurons in zebrafish, as they do in mammals. For example, zebrafish have an extensive histaminergic system with multiple histamine receptors, but histamine receptor expression appears to be restricted to the central nervous system (*Peitsaro et al., 2007*). Furthermore, while zebrafish do possess mast cells (which in mammals release histamine and serotonin that act upon peripheral sensory neurons to elicit itch), studies indicate that these mast cells do not contain histamine or serotonin (*Prykhozhij and Berman, 2014*; *Mulero et al., 2007*). It is therefore plausible that in the zebrafish, histamine and serotonin do not play a functional role in pruritic sensation or immune responses, but are restricted to more central processes such as mediating sleep-wakefulness cycles, mood, and anxiety (*Pei et al., 2016*; *Sackerman et al., 2010*).

The direct activation of a TRP channel to elicit itch or pain through the recruitment of distinct populations of somatosensory neurons via stimulus sensitivity is mechanistically simple. That such a mechanism is present in both zebrafish and mouse may imply that this particular form of itch transduction appeared before the emergence of terrestrial vertebrates, and may have been conserved throughout evolutionary history to persist in mammals. One could therefore postulate that this rudimentary form of itch potentially evolved from existing pain-transducing molecular and neural machinery as a means to differentiate high- and low-intensity noxious stimuli for which an alternative behavioral response was more appropriate. While data from only two species is not sufficient to declare that our proposed mechanism is evolutionarily conserved, we hope that our findings provide a springboard for future research into pruritus across multiple taxa, which would help elucidate the evolutionary development of this important sense.

## Conclusion

Our work demonstrates that IMQ can directly activate TRPA1 to elicit pruritic behavioral responses in both the zebrafish and mouse. Furthermore, we have shown that the immune receptor TLR7 does not mediate somatosensory neuronal responses to IMQ. Our results imply that in both species a subset of highly sensitive TRPA1-expressing itch-encoding neurons can respond to weaker TRPA1 agonists to encode sensations of itch and elicit discrete itch behaviors. More intense stimuli, such as those that evoke nocifensive behaviors, appear to recruit this highly-sensitive subset as well as less-sensitive TRPA1-expressing neurons. Our finding that IMQ responsive neurons in the mouse are part of a TRPA1-expressing subpopulation that is activated by other TRPA1 dependent pruritogens provides further evidence that these more sensitive neurons indeed signal itch. Parallel observations between the zebrafish and mouse suggest that this relatively simple mechanism for conveying, and distinguishing between, pruritic and algogenic stimuli originated early in vertebrate evolution and appears to be preserved in mammals. In sum, our results support the existence of a population-coding based strategy through which differential activation of TRPA1-expressing somatosensory neurons with high or low sensitivities to TRPA1 agonists can relay the discrete sensations of itch and pain respectively.

## Materials and methods

### Zebrafish husbandry

Adult Zebrafish (*Danio rerio*) were raised with constant filtration, temperature control (28.5 ± 2°C), illumination (14 hr:10 hr light-dark cycle, lights on at 9:00 AM), and feeding. All animals were maintained in these standard conditions and the Institutional Animal Care and Use Committee approved all experiments. Adult zebrafish not used in behavioral experiments were bred in spawning traps (Thoren Caging Systems, Hazelton, PA) from which embryos were collected. Embryonic and larval zebrafish were raised in petri dishes (Fisher Scientific, Hampton, NH) of E2 medium with no more than 50 embryos per dish at 28.5 ± 1°C in an incubator (Sanyo). Embryos were staged essentially as described (*Kimmel et al., 1995*) and kept until 5dpf.

### Mouse husbandry

*Trpa1$^{+/+}$* and congenic *Trpa1$^{-/-}$* mice on the *C57BL/6J* background were described previously (*Cruz-Orengo et al., 2008*). All mice were housed under a 12 hr light/dark cycle with food and water provided ad libitum. All behavioral tests were videotaped from a side angle, and behavioral assessments were done by observers blind to the treatments or genotypes of animals. All mice used for behavior tests were age, sex and body weight matched. All experiments were performed in accordance with the guidelines of the National Institutes of Health and the International Association for the Study of Pain, and were approved by the Animal Studies Committee at Washington University School of Medicine.

### Cell lines

HEK 293T cell stocks were initially purchased from ATCC, which authenticated their identity via STR profiling. Cells tested negative for mycoplasma contamination. Cells were cultured in DMEM (Life Technologies, Carlsbad, CA) supplemented with fetal bovine serum and antibiotics (penicillin/streptomycin), and passaged every 2–3 days.

### Genomic DNA extraction

Individual larvae were processed as previously described (*Meeker et al., 2007*). Larvae were anesthetized with tricaine, and placed in individual PCR tubes with a small quantity of E2 media. An equivalent amount of a 2X base solution made from a 50x stock (1.25 M NaOH, 10 mM EDTA pH 12) was then added to each tube, and all tubes were incubated at 95°C for 30 min. Following this, 1x neutralization solution (again made from from a 50x solution, 2M Tris-HCl pH 5) was added, and the resulting DNA solutions were stored at −20°C. Adult genomic DNA was extracted using similar methods, but with a few minor modifications. Individual fish were anesthetized with tricaine, and a small portion of the tail fin was removed with a scalpel and placed into an individual PCR tube. 1X base solution was then applied to the piece of tissue, which was incubated for 30 min at 95°C until an equivalent amount of 1X neutralization solution was added.

### Nonsense mutant generation

Nonsense mutants for both *trpa1b* and *tlr7* were generated essentially as previously described (*Shah et al., 2016*). To synthesize the template DNA required for the in vitro transcription we employed a two oligo PCR method, one oligo contained the RNA loop structure required for recognition by the Cas9 enzyme and had the sequence 5'[gatccgcaccgactcggtgccactttttcaagttgataacggactagccttattttaacttgctatttctagctctaaaac]3'. The second, gene specific, oligo had the sequence 5'[aattaatacgactcactata(N20)gttttagagctagaaatagc]3', where (N20) refers to the 20 nucleotide oligo that binds the genome. In the case of the *trpa1b* nonsense mutants the N20 oligo was 5'[GGCGTA TAAATACATGCCAC]3'. In the case of the *tlr7* nonsense mutant the N20 oligo was 5'[GGGGATG TAGGACAAGTTGT]3'. A mixture of 400 µL of Cas9-encoding mRNA and 200 ng/µL of the proper sgRNA was injected into zebrafish embryos of the AB background at the one cell stage.

Fish were then screened for mutations using the following primers, *tlr7* 5' GGATGCGTTTATGC TGCTTGACAA, *tlr7* 3' AATGTTGTTGTTGTACAGGTAGAGCTC, *trpa1b* 5'-CTCATACATTCA TAAACCTGCCTGATAT, and *trpa1b* 3' – TGGAGGGGCGTCAGACCCTTT, and Sanger Sequencing.

We identified two nonsense mutants for *trpa1b*, one that possessed a 4 bp insertion and one that possessed a 7 bp deletion. We then outcrossed these founders (F0) to WT fish of the same genetic background (AB line) and screened for germline transmission in the F1 generation. Members of the F1 generation were additionally backcrossed to WT fish to establish an F2 generation. Heterozygous F2 zebrafish were then crossed to each other to produce an F3 generation that was used for experiments; additionally F2 zebrafish were crossed to transgenics expressing calcium indicator proteins (CaMPARI, GCaMP) under neuronal promoters in order to perform functional imaging studies. In some instances, F3 zebrafish were backcrossed a fourth time to establish younger generations of fish. Animals that were homozygous for either the 4 bp insertion or 7 bp deletion possessed identical phenotypes (i.e., lack of behavioral response to AITC). Likewise, zebrafish with a 4 bp/7 bp phenotype possessed an identical phenotype as 4 bp/4 bp and 7 bp/7 bp homozygotes.

We identified one nonsense mutant for *tlr7* that possessed a 1 bp deletion. As with the generation of the *trpa1b* mutant line, this founder was outcrossed to a WT fish and the offspring were screened for germline transmission. Subsequent generations were backcrossed in the manner described above.

In the *trpa1b* experiments, WT siblings of *trpa1b*$^{-/-}$ fish were used as the controls. In the *tlr7* experiments, a pure *tlr7*$^{-/-}$ line was established and compared to age-matched WT fish, since we were unable to genotype the 1 bp mutation via conventional methods (gel electrophoresis, HRMA) and could only identify the mutation via sequencing.

## Larval zebrafish behavior

At 5dpf, larval zebrafish (AB background) were placed into individual wells on a 96-well mesh bottom plate (Millipore, Burlington, MA) resting in a bath of E2 medium. The 96-well plate was then transferred to a hot plate that was maintained at a constant temperature of 28.5°C. Then the 96-well plate was moved from the E2 medium bath to the experimental bath for four minutes, during which the behavioral response of the larval zebrafish was recorded with a HD camcorder (Canon, Japan). Experiments were performed blindly and each larva's total locomotive behavioral response was tracked using Ethovision (Noldus, Netherlands). Statistical analysis was done using an analysis of variance (Graphpad Prism 6) or Student's *t*-test. All experimental compounds were purchased from Sigma Aldrich unless otherwise noted and were made up in 1% dimethyl sulfoxide (DMSO, Sigma Aldrich, St. Louis, MO) and E2 medium.

In experiments involving nonsense mutants and their WT siblings, all larvae were genotyped following video capture of the behavioral response. Briefly, each larva was removed from its well and placed into a PCR tube in 25 μL of E2 media. gDNA was extracted using the base extraction technique described above. For experiments involving *trpa1b*$^{-/-}$ fish, all larvae were genotyped by HRMA (CFX Connect, BioRad, Hercules, CA) using the primers 5'-CTCATACATTCATAAACCTGCCTGATAT and 3'-TGGAGGGGCGTCAGACCCTTT. As mentioned previously, due to difficulties in genotyping the 1 bp deletion in *tlr7* nonsense mutants, animals were identified by genomic sequencing, and a pure *tlr7*$^{-/-}$ was created for use in behavioral experiments and were compared to AB fish.

## Examining neuronal activation with CaMPARI transgenic zebrafish

*elavl3*:CaMPARI zebrafish in the Casper background were simultaneously exposed to chemical stimuli and a 405 nm light in order to permanently photoconvert active neurons (*Fosque et al., 2015*). Briefly, 3dpf larval zebrafish were paralysed by injecting α-bungarotoxin protein (Sigma) into the chest cavity using microinjection needles pulled on a Flaming-Brown Micropipette Puller (model P-87, Sutter Instrument Co., Novato, CA) and a Picrosprizter II microinjection apparatus (General Valve Corporation, Fairfield, NJ). Paralysed fish were then placed in small glass-bottomed dishes (Wilco Wells, Netherlands) filled with an individual chemical from the pruritic screen and allowed to incubate for 2 min. Following this incubation period, glass-bottomed dishes were placed on the stage of an inverted fluorescent microscope (Olympus, Japan, model Ix81S1F-3) and the larvae were exposed to a 405 nm light for 40 s using MetaMorph software (Molecular Devices, San Jose, CA). Post-exposure fish were removed from the chemical and placed in a petri dish filled with embryo media and tricaine to prevent any future activation of sensory neurons. Immediately prior to imaging, larvae were mounted on coverslips in 1.5% agarose +tricaine in EM. TG and surrounding neural tissue were imaged using a 20x lens on an LSM 880 confocal microscope (Zeiss, Germany). Zen Black

software was used to scan through the entire TG, acquiring a 1024 × 1024 pixel image slice at every ~5 μm that could then be stacked in the Z plane until the entire ganglion was imaged. Images were examined for photoconverted (red-labeled) neurons, and totals were established for each TG in each condition. When used, ANOVA statistical tests were done against control.

## Adult zebrafish behavior

Adult zebrafish were placed in traps (Thoren Caging Systems) and were transported to the experimental area, which was maintained at 28.5 ± 2°C, where they were left to acclimate for one hour. After completing acclimation fish were transported one at a time to the injection area. Each fish was anaesthetized by exposure to 12.0 ± 0.3°C system water. They were then immobilized and injected in the upper lip using a 33 gauge Hamilton needle and 20 μL Hamilton syringe. Fish were injected with 10 μL of experimental or control solution, all of which were made up in 1% DMSO, 1x PBS, and distilled water. After injection, fish were placed into a trap and transferred to the recording area. The behavioral response was recorded for five minutes using an HD camcorder (Cannon). The velocity of the fish was then analyzed using Ethovision (Noldus) and all facial interactions were manually scored to prevent any bias in the data. All analysis was blinded. All statistical analysis were done with an analysis of variance (Graphpad Prism 6) or Student's t-test.

In the case of nonsense mutant experiments, after the behavioral responses were captured, each fish was euthanized by tricaine overdose and fin clipped, and the fin section was placed in a PCR tube. Then, gDNA was extracted from the excised tissue using the previously described base extraction technique. *Trpa1b* genotype was determined using the same HRMA strategy as employed in the larval behavioral experiments. Since a homozygous $tlr^{-/-}$ line was employed for adult behavioral experiments, genotyping post-experiment was unnecessary. When used, ANOVA statistical tests were done against control.

## Larval *in situ* hybridization experiments (zebrafish)

Whole-mount colorimetric *in situ* hybridization to determine *trpa1b* and *tlr7* expression was performed on 3dpf larvae as described previously (*Gau et al., 2013*). Pigment formation was inhibited by exposing larvae to 1-phenyl 2-thiourea (PTU) at 24hpf. Larvae were hybridized with DIG-labeled riboprobes for *trpa1b* or *tlr7* overnight at 65°C. They then underwent a series of stringent washes, followed by incubation in α-DIG conjugated Fab fragments (Roche, Switzerland, 1:10,000) and staining in NBT/BCIP solution. Larvae were washed with PBTw and stored in glycerol until imaging, whereupon they were mounted in 100% glycerol and photographed using an upright Axioplan2 microscope (Zeiss).

## Calcium imaging with *elavl3*:GCaMP transgenic zebrafish

3dpf zebrafish larvae from either *elavl3*:GCaMP5 (*Akerboom et al., 2012*) or *elavl3*:H2BGCaMP6 (*Chen et al., 2013*) transgenic line were paralysed as described above. After paralysis, larvae were mounted in 2% agarose in EM on coverslips, which were then placed into a perfusion chamber (Warner Instruments). Once solidified, the agarose immediately surrounding the head was cut away with a scalpel to ensure maximal exposure to chemical stimuli. The perfusion chamber was placed onto the stage of an Olympus Fluoview FV-1000 multiphoton microscope equipped with an infrared laser controlled by Mai Tai software (Spectra-Physics, Thermo Electron Corporation, Walthom, MA). Larvae were imaged under the following parameters: laser wavelength of 880 nm, resolution of 4.0 μs/pixel, frame rate of 1–3 s per frame, frame size of 512 × 512 pixels. Laser intensity, HV, and zoom were optimized for individual larva. For experiments comparing multiple stimuli, each stimulus was separated by an equivalent period of E2 media washout. All solutions were made in E2 media containing 2% DMSO.

For calcium imaging experiments involving $trpa1b^{-/-}$ animals and their WT/$trpa1b^{+/-}$ siblings, *elavl3*:GCaMP5:$trpa1b^{-/-}$ larvae were employed. One day prior to imaging, larvae were anesthetized with tricaine, tail-clipped, genotyped via HRMA, and housed in individual wells within a 24-well plate until ready for use in experiments. 2-APB was used a positive control.

## Ratiometric calcium imaging

DRGs were isolated from 6- to 12-week-old C57Bl/6J mice. All experiments were performed in compliance with institutional animal care and use committee standards and experiments were performed essentially as described (*Kimball et al., 2015*). Dissociation and culturing of mouse DRG neurons were performed as described with the following modifications (*Story et al., 2003*). Dissected DRGs were dissociated by incubation for 1 hr at 37°C in a solution of culture medium [Ham's F12/Dulbecco's modified Eagle's medium (DMEM) with 10% horse serum, 1% penicillin-streptomycin (Life Technologies, Carlsbad, CA)] containing 0.125% collagenase (Worthington Biochemicals, Lakewood, NJ), followed by a 30 min incubation in 10 ml of culture media plus 1.25 units of papain. Calcium imaging was performed essentially as described previously (*Story et al., 2003*). Growth media was supplemented with 100 ng/ml nerve growth factor. For experiments involving heterologous expression, human embryonic kidney (HEK) 293T cells were transiently transfected with one or two of the following plasmid constructs: zebrafish *trpa1b*, zebrafish *tlr7*, mouse *Trpa1*, mouse *Tlr7*, human *TRPA1*, and/or human *TLR7*. All constructs except for the one encoding zebrafish *tlr7* were also co-transfected with pIRES-eGFP plasmid in order to estimate transfection efficiency. (For zebrafish *tlr7*, this step was unnecessary because the construct was already in the pIRES-eGFP vector.) The buffer solution for all experiments was 10 mM HEPES in 1X Hanks' balanced salt solution (HBSS) (Invitrogen, Carlsbad, CA).

The threshold for activation was defined as 30% above baseline for both DRG and heterologous expression experiments. Student's *t*-test was used for all statistical calculations. All averaged traces represent mean ± s.e.m. All reported fluorescence values of each cell were normalized to the fluorescence of that cell during the initial baseline wash period. Maximum response values of each cell were calculated as the difference between the maximum and minimum fluorescence values of the cell during a stimulus application period.

## Dual luciferase assay

To verify the functionality of our transfected *Tlr7* constructs, we determined levels of NF-kB induction following stimulation with the TLR7 agonist loxoribine using a Dual-Luciferase Reporter Assay System (Promega, Madison, WI). Briefly, HEK 293T cells were seeded at ~80% confluency in individual wells of a 24-well plate (N $\approx$ 4 $\times$ $10^5$ cells per well) and transiently transfected with the same zebrafish, mouse, or human *Tlr7* constructs as employed in calcium imaging following a standard lipofectamine protocol. All cells were also co-transfected with a nF-kB Firefly luciferase reporter plasmid (p1242 3x-KB-L, Addgene [*Mitchell and Sugden, 1995*]), a Renilla luciferase control plasmid (p207-CMV-Renilla, gift from Tom Reh), and pIRES-eGFP (if necessary) for estimating transfection efficiency. As a negative control, another set of HEK 293T cells were transfected only with pIRES-eGFP. Twenty-four hours following transfection, the culture media was removed from all cells and replaced with normal serum-free media or with serum-free media containing 200 µM loxoribine. Following a 24 hr treatment period, culture media was removed, plated cells were rinsed briefly with DPBS, then lysed with passive lysis buffer (Promega) and gentle agitation on a multi-purpose rotator (Barnstead, Hampton, NH). Lysates were analyzed on a Viktor3 1420 Multilabel Counter (PerkinElmer, Waltham, MA), which generated luminescence values in CPS (counts per second) for both Firefly and Renilla luciferase activity. Assays were also performed on 1X PLB samples to estimate background luminescence. Background luminescence was subtracted from each measurement, and the Firefly/Renilla CPS ratio was calculated for each condition.

## Immunohistochemistry on HEK 293T cells

To verify that TLR7 was being expressed by HEK 293T cells used in our experiments, we transfected cells on coverslips with either mouse or human *Tlr7* constructs and pIRES-eGFP; some cells were only transfected with pIRES-eGFP to serve as a negative control. 48 hr following transfection, cell culture media was removed, and coverslips were washed briefly with 1X DPBS and fixed for 10 min at room temperature in 4% paraformaldehyde in 1X PBS (Electron Microscopy Sciences, Hatfield, PA). Coverslips were again rinsed briefly in DBPS and then blocked in 10% goat serum in 1X PBST (PBS with 0.1% Tween-20) for 1 hr at room temperature. Primary antibodies against TLR7 (rabbit anti-TLR7, Boster, Pleasanton, CA, 1:250) and GFP (chick anti-GFP, 1:1000, Invitrogen) were made in PBST with 10% goat serum and applied to the coverslips, which were

incubated overnight at 4°C. Coverslips were then rinsed 3X in PBST to remove primary antibodies, treated with secondary antibodies (AlexaFluor goat anti-chicken 488 and AlexaFluor goat anti-rabbit 568, both at 1:1000, Life Technologies and Invitrogen, respectively) for approximately 2 hr at room temperature, washed in PBST, and mounted on slides with DAPI-containing Vectashield medium (Vector Laboratories, Inc., Burlingame, CA). Confocal imaging of mounted cells was performed using a Zeiss microscope and Zen Black acquisition software.

## Electrophysiology

Whole-cell patch-clamp recordings were performed at room temperature (22–24°C) using an Axon 700B amplifier (Molecular Devices, Sunnyvale, CA) on the stage of an inverted phase-contrast microscope equipped with a filter set for GFP visualization (Nikon Instruments Inc., Melville, NY, USA) (*Feng et al., 2017*). Pipettes pulled from borosilicate glass (BF 150-86-10; Sutter Instrument, Novato, CA) with a Sutter P-1000 pipette puller had resistances of 2–4 for whole-cell patch-clamp recordings when filled with pipette solution containing 140 mM CsCl, 2 mM EGTA, and 10 mM HEPES with pH 7.3 and 315 mOsm/l osmolarity. Cells were perfused with extracellular solution containing 140 mM NaCl, 5 mM KCl, 0.5 mM EGTA, 1 mM $MgCl_2$, 2 mM $CaCl_2$, 10 mM glucose, and 10 mM HEPES (pH was adjusted to 7.4 with NaOH, and the osmolarity was adjusted to $\approx$ 340 mOsm/l with sucrose). The whole-cell membrane currents were recorded using voltage ramps from $-100$ to +100 mV for 500 ms at holding potential of 0 mV. Data were acquired using Clampex 10.4 software (Molecular Devices, Novato, CA). Currents were filtered at 2 kHz and digitized at 10 kHz. Data were analyzed and plotted using Clampfit 10 (Molecular Devices, Novato, CA). The concentration-response curve was fitted with the logistic equation: $Y = Ymin + (Ymax - Ymin)/(1 + 10[(logEC_{50} - X) \times Hill\ slope])$, where Y is the response at a given concentration, Ymax and Ymin are the maximum and minimum responses, X is the logarithmic value of the concentration and Hill slope is the slope factor of the curve. $EC_{50}$ is the concentration that gives a response halfway between Ymax and Ymin. All data are presented as mean $\pm$ s.e.m.

## Mouse behavior

Mice were shaved on the nape of the neck or on the face two days before the assay. On the day of experiment, mice were acclimated for 1 hr by placing each of them individually in the recording chamber followed by intradermal injection of 10 µg of IMQ to the nape of the neck. Immediately after the injection, mice were videotaped for 30 min without any person in the recording room. After the recording, the videotapes were played back and the number of scratching bouts towards the injection site was counted by an investigator blinded to the treatment.

Cheek injection of AITC was performed as described (*Shimada and LaMotte, 2008*). Briefly, during anesthesia with isoflurane (2% in 100% oxygen), the right cheek (approx. 5 × 8 mm area) was shaved. Mice were acclimated in the recording chambers at least two days before experiments began. AITC was dissolved in DMSO to make a 5 M stock solution and diluted in saline to make the working solution. Every mouse received an injection of 10 ul volume at the shaved area. Immediately after the injection, mice were videotaped for 30 min without any person in the recording room. After the recording, the videotapes were played back and the number of scratching and wiping bouts towards the injection site was counted by an investigator blinded to the treatment.

Footpad injections were performed as described previously (*Liu et al., 2016*). Briefly, either a saline control or IMQ (125 µg) was injected into the footpad of the hindpaw. Animals were filmed, and nocifensive behaviors (licking, biting) in the recorded videos were scored by an investigator blind to the treatment.

## Acknowledgements

This work was primarily supported by NIH R01DE23730 (AD). Other support includes the Levinson Emerging Scholars award (LC) and a Mary Gates Undergraduate Research Award (LC). We thank David Raible and Hongzhen Hu for extensive feedback on experimental design and the manuscript, Paul Nakamura and Quynh Nguyen for technical support with the dual luciferase assay, Tom Reh for the p207-CMV-Renilla luciferase plasmid, Rachel Wong for use of the FluoView 1000 multiphoton microscope, and David White for zebrafish husbandry support.

## Additional information

### Funding

| Funder | Grant reference number | Author |
| --- | --- | --- |
| National Institutes of Health | R01DE23730 | Ajay Dhaka |
| Mary Gates | Undergraduate Research Research Award | Logan Condon |
| Levinson Emerging Scholars Award | Undergraduate Research Award | Logan Condon |

The funders had no role in study design, data collection and interpretation, or the decision to submit the work for publication.

### Author contributions

Kali Esancy, Logan Condon, Jing Feng, Data curation, Formal analysis, Validation, Investigation, Visualization, Methodology, Writing—original draft, Writing—review and editing; Corinna Kimball, Andrew Curtright, Data curation, Formal analysis, Validation, Investigation, Methodology; Ajay Dhaka, Conceptualization, Resources, Supervision, Funding acquisition, Visualization, Methodology, Writing—original draft, Project administration, Writing—review and editing

### Author ORCIDs

Ajay Dhaka http://orcid.org/0000-0001-5783-8582

### Ethics

Animal experimentation: Experiments using zebrafish were performed under the University of Washington Institutional Animal Care and Use Committee protocols #4216-02 (approved on 9/16/2016). The University of Washington Institutional Animal Care and Use Committee (IACUC) follow the guidelines of the Office of Laboratory Animal Welfare and set its policies according to The Guide for the Care and Use of Laboratory Animals. The University of Washington maintains full accreditation from the Association for Assessment and Accreditation of Laboratory Animal Care (AAALAC) and has letters of assurance on file with OLAW. The IACUC routinely evaluates the University of Washington animal facilities and programs to assure compliance with federal, state, local, and institution laws, regulations, and policies. The OLAW Assurance number is DL16-00292.

### Decision letter and Author response

Decision letter https://doi.org/10.7554/eLife.32036.024
Author response https://doi.org/10.7554/eLife.32036.025

## Additional files

### Supplementary files

• Transparent reporting form
DOI: https://doi.org/10.7554/eLife.32036.018

### Major datasets

The following previously published datasets were used:

| Author(s) | Year | Dataset title | Dataset URL | Database, license, and accessibility information |
| --- | --- | --- | --- | --- |
| Howe K | 2016 | transient receptor potential cation channel, subfamily A, member 1b | https://www.ncbi.nlm.nih.gov/gene/474353 | NC_007135.7 |

| Howe K | 2016 | toll-like receptor 7 | https://www.ncbi.nlm.nih.gov/nuccore/NC_007120.6?report=genbank&from=54570602&to=54581245 | NC_007120 |

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
