## [Decision Letter]

Thank you for submitting your article "An evolutionarily conserved model for selective pruritus via direct activation of TRPA1" for consideration by *eLife*. Your article has been reviewed by three peer reviewers, and the evaluation has been overseen by Gary Westbrook as the Senior Editor and a Reviewing Editor. The following individual involved in review of your submission has agreed to reveal her identity: Diana Bautista (Reviewer #1). The reviewers have discussed the reviews with one another and the Editor has drafted this decision. The reviewers were interested in the topic, but had a number of concerns, some more serious than others, and some of which will require additional experiments.

Summary:

This is an interesting and thoughtful study examining itch in zebrafish. The authors first performed a screen to identify pruritogens that activate the zebrafish trigeminal ganglia and show that imiquimod triggers calcium influx in these neurons. They also show that imiquimod injection into the lip triggers a novel behavioral response, namely lip rubbing. TRPA1, but not TLR7, mediated the cellular and behavioral responses to imiquimod, and zebrafish, mouse and human TRPA1 channels were directly activated by imiquimod. However, Trpa1 is expressed in a large number of nociceptive neurons and produces pain as well as itch responses. The authors propose that imiquimod and AITC differentially activate subsets of Trpa1-expressing neurons such that at low concentrations it produces itch occurs with low concentrations and nociception at high concentrations.

Essential revisions:

1) It is not clear how many pruritogens were tested in the CaMPARI screen. The authors state "a variety" and the figure shows some data. Please clarify. In general the screen could be explained better. Most readers will not have heard of CaMPARI, so a brief explanation of how it works would improve access for the reader.

2) Some additional information about the lip rubbing behavior is needed. How long does the behavior persist? Figure 1K shows representative responses to 10uM AITC and 10uM imiquimod. The responses to 100 and 200uM imiquimod and also low dose AITC should be displayed as this data best captures the differences between the behaviors.

3) The authors state that the lack of effect of loxoribine on neuronal activation and behavior is similar to "reports in the mouse that loxoribine does not elicit pruritic behavioral responses (Liu et al., 2010)". Liu et al. state that loxoribine does induce a dose-dependent, TLR7-dependent itch in mice. But their actual data show that scratching is very transient and only statistically significant for a first few minutes following injection. Were loxorobine-evoked behavioral and calcium imaging experiments performed such that a rapid response would be detected? If so, please report the response, or lack of response, during the first few minutes following injection.

4) It is interesting that mouse and human TLR7 is activated by both imiquimod and loxoribine but that fish TLR7 is only sensitive to imiquimod. Are there other differences in the zebrafish response to TLR7 agonists, such as the natural ligand ssRNA? This is worthy of discussion given that the mechanisms are unknown by which parasite infection of fish triggers natural behaviors to remove the parasite (e.g., visiting cleaning stations, rubbing against coral). TLRs are proposed to be one mechanism by which fish respond to parasite infection, but few studies have addressed this topic in detail. A short discussion of previous work on infection, natural fish behaviors and TLRs would expand the scope of the study.

5) A general mechanism for discrimination of itch from pain is proposed. Although the evidence is mostly consistent for the proposed (see, my major reservations below), it was not convincing that this is the principle mechanism by which most itch agents are discriminated from nociceptive stimuli. Experiments showing that other pruritogens are discriminated via the same proposed mechanism would be needed. At the least, discussion of alternative mechanisms is necessary.

6) Assuming zebrafish have mast cells, the authors should discuss why histamine and serotonin do not elicit itch-behavior in fish. Also, data from two species does not seem to be enough to claim that the mechanism is evolutionary conserved. This claim should be toned down.

7) The behavioral experiments performed in fish, with different doses of AITC need to be performed in mice, i.e. the proposed mechanism would predict that low doses of AITC should elicit scratching in mice, while higher doses would provoke nociferous responses. This is not a difficult experiment and is a test of the authors' proposed mechanism.

8) The data that the zebrafish Trpa1 is the imiquimod receptor in vivo and in vitro is strong, however the behavioral results for mouse Trpa1 is weak. First, at the doses of IMQ used in mice, only small numbers of scratch-responses are elicited. Second, Trpa1-null mice, compared to wild-type mice, only showed a 50% reduction in responses (not the same as the almost complete loss seen in fish). What is the result at higher doses of IMQ in mice? The presented results suggest that there are additional IMQ receptors in mice that contribute to scratching responses.

9) The authors suggest that selective subsets of Trpa1-neurons are activated, but provide little direct evidence for this proposal. The authors should investigate if other known mouse itch-inducing agents activate the same neurons as IMQ. This experiment could be performed relatively quickly.

10) More information about the CRISPR mutants and additional validation would be appreciated. How many times were the mutants backcrossed? Were multiple alleles identified, and did they have the same phenotype? Can the mutant phenotype be rescued by expressing wild type TrpA1b? In most experiments wt was used as the control; why weren't heterozygous siblings used?

11) Is the IMQ-responsive subset of neurons in larvae genetically specified or determined stochastically? This could be addressed by imaging responses to IMQ at different times and determining if the same neurons are activated each time.

12) These experiments show that IMQ responses require TrpA1b, but not TLR7. They also show that in heterologous cells, expression of TrpA1b, but not TLR7, confers a response to IMQ, albeit more weakly than AITC. These results lead the authors to propose that TrpA1b is the direct sensor of IMQ, rather than a channel downstream of a GPCR. The fact that TrpA1b can be directly activated is compelling, but the results do not exclude the possibility that there is another receptor in vivo. This issue should be discussed.

13) The fact that low concentrations of AITC activate a similar fraction of TrpA1b-expressing neurons in zebrafish larvae as IMQ is a nice finding supporting the authors' hypothesis that IMQ-expressing neurons are primed for TrpA1 activation. This idea was strengthened by an experiment showing that IMQ-responsive DRG neurons are enriched in the population of neurons activated by low AITC concentrations. This finding could be strengthened further by addressing the same question with calcium imaging in zebrafish larvae- i.e. are the same trigeminal neurons that respond to IMQ also those responsive to low TrpA1 concentrations?

---

## [Author Response]

Essential revisions:1) It is not clear how many pruritogens were tested in the CaMPARI screen. The authors state "a variety" and the figure shows some data. Please clarify. In general the screen could be explained better. Most readers will not have heard of CaMPARI, so a brief explanation of how it works would improve access for the reader.

We agree that our previous description of the screen was vague and have updated that particular section in the main text. We screened five drugs that were selected based on their site of action. We exclusively screened drugs known to act on mammalian receptors with zebrafish orthologs, which limited the number of compounds we could test. For example, we did not test pruritogens known to act upon Mrgprs, which are not present in the zebrafish. Additionally, to increase reader accessibility we added a description of the CaMPARI protein.

“In an effort to determine if pruritic stimuli are capable of eliciting somatosensory activity in zebrafish, we screened five compounds known to both induce acute pruritus in mammals and act on receptors expressed by zebrafish (Schön and Schön, 2007; Bell, McQueen and Rees, 2004; Lieu et al., 2014; Yaaguchi et al., 1999; Tsujii et al., 2009), excluding pruritogens that act on receptors that do not have a zebrafish ortholog, such as MRGPR agonists. […] Using this approach, we were able to view trigeminal neuronal activity in 3 day post fertilization (dpf) larval zebrafish following the application of each pruritogen (Figure 1A-H).”

2) Some additional information about the lip rubbing behavior is needed. How long does the behavior persist? Figure 1K shows representative responses to 10uM AITC and 10uM imiquimod. The responses to 100 and 200uM imiquimod and also low dose AITC should be displayed as this data best captures the differences between the behaviors.

When initially developing the adult behavior assay we determined that the vast majority of lip rubbing behavior occurred during the first five minutes post injection. As a result, this time was utilized for all further behavioral quantifications with adult zebrafish. We would also like to note that Figure 1K was mislabeled. The displayed responses are from Control, 10uM AITC, and 100uM IMQ respectively. We have also added Control, 5 µM AITC, 10 µM AITC, 100 µM IMQ, and 200 µM IMQ behavioral traces to Figure 4—figure supplement 1.

3) The authors state that the lack of effect of loxoribine on neuronal activation and behavior is similar to "reports in the mouse that loxoribine does not elicit pruritic behavioral responses (Liu et al., 2010)". Liu et al. state that loxoribine does induce a dose-dependent, TLR7-dependent itch in mice. But their actual data show that scratching is very transient and only statistically significant for a first few minutes following injection. Were loxorobine-evoked behavioral and calcium imaging experiments performed such that a rapid response would be detected? If so, please report the response, or lack of response, during the first few minutes following injection.

We thank the reviewers for identifying our mislabeled citation. The sentence in the text was intended to cite Kim et al., 2011 (not Liu et al., 2010), and has since been corrected. Kim et al. (Kim et al., 2011) performed similar experiments as Liu et al., but did not find any increase in scratching behavior when mice were injected with loxoribine, in line with what we observed in the zebrafish. However, the reviewer addresses an important issue regarding the time course of behavioral experiments. According to Liu et al., loxoribine only evokes a statistically significant increase in mouse scratching behavior within the first five minutes following injection. In all of our behavioral experiments with adult zebrafish, fish were filmed immediately after injection for 5 minutes. Therefore, we would assume that our behavioral assay was temporally sensitive enough to capture even transient, rapid responses to loxoribine. That we did not observe any immediate responses to loxoribine indicates that it likely has no effect on zebrafish. Larval zebrafish behavior was conducted on similar timescale, as recording began immediately after the mesh-bottomed plate containing zebrafish larvae was lowered into the chemical to be tested. We would therefore expect that our larval behavioral screen has the capacity to capture brief, transient responses to loxoribine. While we did not performin vivo GCaMP calcium imaging on larval zebrafish with loxoribine, our CaMPARI experiments would be able to capture immediate neuronal responses. In these experiments, photoconversion was performed while the larvae were still in the stimulus bath; as such, there was no temporal “gap” between stimulus exposure and photoconversion of active neurons. Given a lack of response to loxoribine under all of these experimental conditions, especially when coupled with our data from the dual luciferase assay indicating that loxoribine did not activate zebrafish TLR7, we concluded that loxoribine has no behavioral or neuronal effects on the zebrafish.

While we intended for our reference to the Kim et al. paper (Kim et al., 2011) to highlight the similarities of our results in the zebrafish with previous findings in the mouse, and not to necessarily compare our results in mice to these previous findings, it is of note that our calcium imaging experiments with dissociated mouse DRGs would also be sufficient to capture a transient loxoribine response. All of our DRG calcium imaging experiments were performed in real time – i.e., neurons were imaged while chemicals were being perfused across them – and thus we would expect that any immediate responses could be captured. We did not perform any loxoribine behavioral experiments in mice, and so cannot comment on the response/lack thereof following injection.

It should also be noted that the experiments performed by Kim et al. were binned in 5 minute increments, like the experiments performed by Liu et al. Therefore, we would expect that their assay had identical temporal sensitivity, and would have been able to capture the transient loxoribine responses Liu et al. observed.

4) It is interesting that mouse and human TLR7 is activated by both imiquimod and loxoribine but that fish TLR7 is only sensitive to imiquimod. Are there other differences in the zebrafish response to TLR7 agonists, such as the natural ligand ssRNA? This is worthy of discussion given that the mechanisms are unknown by which parasite infection of fish triggers natural behaviors to remove the parasite (e.g., visiting cleaning stations, rubbing against coral). TLRs are proposed to be one mechanism by which fish respond to parasite infection, but few studies have addressed this topic in detail. A short discussion of previous work on infection, natural fish behaviors and TLRs would expand the scope of the study.

We thank the reviewer for the suggestion that we more thoroughly probe this topic, and as requested, we have expanded upon our discussion of what is known about TLRs and immune responses to infection in teleost fish. Examining the indirect role of TLR7 at promoting endogenous itch in fish would be a very interesting basis for future studies, but was outside the scope of ours. While we found several studies that examined that implicated TLRs in innate immune responses in several fish species (for example, TLRs are upregulated following infection, and TLR activation is associated with proliferation of peripheral blood leukocytes, inhibition of viral replication, and the induction of interferon genes), we were unable to find studies that focused specifically on the link between natural fish behaviors and immune stimulation. We hope that our modifications (Discussion, ninth paragraph) adequately address the reviewer’s concerns. Additionally, this suggestion has brought to light that we could have been more clear in explaining the results of the dual luciferase assay, as we found that zebrafish TLR7 did not respond to either imiquimod or loxoribine, while both human and mouse TLR7 did. We have since rephrased our text to emphasize that zebrafish TLR7 does not respond to these TLR7 agonists. However, we recognize that due to a lack of a positive control, we cannot verify that zebrafish TLR7 was indeed functional or expressed properly in our assay, and have elaborated upon this caveat in the third paragraph of the subsection “Imiquimod directly activates TRPA1”.

“In the zebrafish, however, TLR7 is unlikely to contribute even indirectly to IMQ-induced itch sensation, as we observed no behavioral phenotype in TLR7-/- animals, and found no evidence that TLR7 is even responsive to typical mammalian TLR7 agonists, like loxoribine and IMQ. […] While our results suggest that TLR7 is not involved in transducing IMQ-evoked itch in the zebrafish, investigating the role of immune responses in mediating somatosensations (both in zebrafish and other species) would be an interesting route for future exploration.”

“However, due to a lack of a TLR7 positive control, we were unable to confirm that TLR7 was functional in our heterologous expression system. With this caveat in mind, the lack of zTLR7 response to these TLR7 agonists lends further credence to the conclusion that TLR7 is not involved in somatosensory neuronal activation or behavior in this species.”

5) A general mechanism for discrimination of itch from pain is proposed. Although the evidence is mostly consistent for the proposed (see, my major reservations below), it was not convincing that this is the principle mechanism by which most itch agents are discriminated from nociceptive stimuli. Experiments showing that other pruritogens are discriminated via the same proposed mechanism would be needed. At the least, discussion of alternative mechanisms is necessary.

We thank the reviewer for their comment. To recap, the mechanism we posit for the discrimination of itch from pain is a population coding model in which low intensity TRPA1 agonists activate a selective subset of highly sensitive TRPA1-expressing neurons and high intensity TRPA1 agonists activates both this selective subset as well as another population of less-sensitive neurons. This bears some similarity to “labeled line” coding mechanisms proposed by others, in which a subset of neurons that express pruritic GPCRs and itch-specific neuropeptides convey itch (Bautista, Wilson and Hoon, 2014; Hoon, 2015; Ross, 2011; Goswami et al., 2014; Ma, 2010). In the context of somatosensations provoked by direct TRPA1 channel agonists, we feel that our evidence largely supports such a model, as opposed to other mechanisms (i.e., an intensity coding strategy). As we did not investigate other compounds known to elicit both itch and pain depending upon factors such as stimulus concentration and method of application (Hoon, 2015), however, we cannot claim that this is the universal mechanism through which itch and pain are distinguished. We did not mean to imply that our proposed mechanism was the only method for itch and pain discrimination, and we hope that our modifications to the text have clarified this point. We have amended our Discussion section to emphasize the point that our mechanism may not be applicable to every scenario, and have made efforts throughout the paper to specify that our mechanism refers to TRPA1-expressing neurons being activated by TRPA1 agonists.

“We however do not mean to imply that other pruritogens selectively evoke itch or pain by activating pruriceptors and/or nociceptors in a dose-dependent manner. Our proposed model clearly is not the sole mechanism by which itch is coded as a distinct sensation from pain (Bautista, Wilson and Hoon, 2014; Hoon, 2015; Ross, 2011; Goswami et al., 2014; Ma, 2010).”

6) Assuming zebrafish have mast cells, the authors should discuss why histamine and serotonin do not elicit itch-behavior in fish. Also, data from two species does not seem to be enough to claim that the mechanism is evolutionary conserved. This claim should be toned down.

We thank the reviewer for their thought-provoking question, and have followed through upon the reviewer’s suggestion to discuss this topic more thoroughly. We have expanded our previous few sentences in the Discussion to include a more detailed explanation of zebrafish mast cell physiology (e.g., describing that zebrafish mast cells do not appear to contain histamine or serotonin) and the implications that this may have on itch behavior. Additionally, we have also emphasized the possibility that while zebrafish possess serotonin and histamine receptors, these receptors may not be expressed on somatosensory neurons, precluding their involvement in pruritic sensation (Discussion, eleventh paragraph). The reviewer also points out that data from two species is insufficient to declare that a mechanism is “evolutionarily conserved”. We agree, and have toned down our claims accordingly (Discussion, last paragraph).

Mast Cells: “We observed that out of the several mammalian pruritogens, only IMQ elicited behavioral and neuronal responses in zebrafish. […] It is therefore plausible that in the zebrafish, histamine and serotonin do not play a functional role in pruritic sensation or immune responses, but are restricted to more central processes such as mediating sleep-wakefulness cycles, mood, and anxiety (Pei et al., 2016; Sackerman, 2010).”

Conclusion: “The direct activation of a TRP channel to elicit itch or pain through the recruitment of distinct populations of somatosensory neurons via stimulus sensitivity is mechanistically simple. […] While data from only two species is not sufficient to declare that our proposed mechanism is evolutionarily conserved, we hope that our findings provide a springboard for future research into pruritus across multiple taxa, which would help elucidate the evolutionary development of this important sense.”

7) The behavioral experiments performed in fish, with different doses of AITC need to be performed in mice, i.e. the proposed mechanism would predict that low doses of AITC should elicit scratching in mice, while higher doses would provoke nociferous responses. This is not a difficult experiment and is a test of the authors' proposed mechanism.

We agree that the experiments proposed by the reviewer (i.e. administering different doses of AITC to mice, with the prediction that lower doses should elicit scratching, while higher doses should elicit nocifensive wiping behaviors) are a necessary test of our proposed mechanism. We have since performed these experiments, and our findings demonstrate that low dose AITC (50 mM) produces significant scratching behavior with no appreciable wiping behavior (p < 0.001 and N.S. respectively), while a stronger AITC stimulus (100 mM) results in a significant attenuation of the observed scratching behavior, as well as a significant wiping behavior (p < 0.05 and p < 0.001 respectively). These results have been included in Figure 5..It is not clear why we observed scratch-specific behaviors at AITC concentrations higher than those reported by others to predominantly elicit nocifensive wiping behaviors (Robertson et al., 2013; Akiyama, Carstens and Carstens, 2010).

“Finally, in accordance with our zebrafish data, we observed that AITC stimulus intensity dictates whether mice exhibit pruritic or nocifensive behaviors. […] We observed that AITC (50 mM) produces significant scratching behavior with no appreciable wiping behavior (p < 0.001 and N.S. respectively), while a higher dose of AITC (100 mM) results in a significant attenuation of the observed scratching behavior, as well as a significant wiping behavior (p < 0.05 and p < 0.001 respectively) (Figure 5I).”

8) The data that the zebrafish Trpa1 is the imiquimod receptor in vivo and in vitro is strong, however the behavioral results for mouse Trpa1 is weak. First, at the doses of IMQ used in mice, only small numbers of scratch-responses are elicited. Second, Trpa1-null mice, compared to wild-type mice, only showed a 50% reduction in responses (not the same as the almost complete loss seen in fish). What is the result at higher doses of IMQ in mice? The presented results suggest that there are additional IMQ receptors in mice that contribute to scratching responses.

We agree with the reviewer that the possibility of additional IMQ receptors, and a non-TRPA1 dependent mechanism for mediating IMQ-induced itch, is present in the mouse. To explore this possibility further, we performed experiments in which a higher dose of IMQ (50 μg) was applied to both WT and TRPA1^-/-^ mice. At this concentration we observed increased scratching behavior in wild type animals that did not differ from TRPA1 ^-/-^ mice. These data have been included in Figure 5—figure supplement 1. Higher concentrations of IMQ likely produce this result via an indirect mechanism, potentially through a TLR7 mediated pathway, a possibility that is addressed in our reply to Essential revision #12. We find it unlikely that an alternate pruritic GPCR exists given the overlap we observed between neurons activated by IMQ and low concentrations of AITC, which is also discussed further in our reply to Essential revision #12. Additionally, we would like to clarify that we observed an ~8-fold increase in scratch behavior with IMQ (10 μg) compared to controls, and would argue that this is not a small response.

9) The authors suggest that selective subsets of Trpa1-neurons are activated, but provide little direct evidence for this proposal. The authors should investigate if other known mouse itch-inducing agents activate the same neurons as IMQ. This experiment could be performed relatively quickly.

Although it might not have been clear, we performed calcium imaging experiments that showed that the vast majority of IMQ responsive mouse DRG neurons also responded to the TRPA1 dependent pruritogens chloroquine and deoxycholic acid, implying that these neurons likely comprise a pruritic subset of TRPA1-expressing neurons (Figure 5E, F). We have expanded our explanation of these experiments in the Discussion section, and have likewise clarified how equivalent experiments are impossible in the zebrafish due to a lack of responses to other pruritogens. Our findings that subsequent pulses of IMQ activate the same neurons in a non-stochastic manner (Figure 4—figure supplement 2, see also our response to Essential revision #11), further support our claim that this is a defined subpopulation of TRPA1 neurons.

“Furthermore, in the mouse, we observed that the vast majority of IMQ (100 μM) responsive neurons also responded to the TRPA1-dependent pruritogens chloroquine and deoxycholic acid, supporting the conclusion that these neurons constitute a pruritic subpopulation. Unfortunately, equivalent experiments could not be performed in the zebrafish, as we were unable to identify other pruritogens for this species (Zhang, 2014).”

10) More information about the CRISPR mutants and additional validation would be appreciated. How many times were the mutants backcrossed? Were multiple alleles identified, and did they have the same phenotype? Can the mutant phenotype be rescued by expressing wild type TrpA1b? In most experiments wt was used as the control; why weren't heterozygous siblings used?

We appreciate the reviewer’s concern about the generation and validation of our CRISPR mutants, and have provided more information in the Materials and methods section of our paper. It should be noted that in all behavioral experiments, fish were genotyped after completion of the experiment. While we did not find any evidence of gene dosage effects – i.e., we found no significant difference in AITC response between WT and heterozygous siblings – we restricted our comparisons to wildtype siblings because that seemed to be the most stringent approach. These results are comparable to those obtained by Prober et al. (Prober et al., 2008), who used the TILLING approach to generate mutants. In their study, they observed normal AITC responses in both TRPA1b homozygous wildtype fish as well as their heterozygous siblings. When generating these mutants, we did not perform experiments exploring whether the mutant phenotype can be rescued by expressing wildtype TRPA1b due to technical constraints. We do not have a way to effectively re-institute TRPA1 expression specifically in TRPA1 neurons. While injecting TRPA1 mRNA into a zebrafish embryo would initially seem like a viable procedure, we could not ensure specificity of expression. Additionally, given that mRNA injections are performed early in development (i.e., at the 1-cell stage), the mRNA expression would not last until 5dpf, the age at which we performed all of our behavioral screens due to mRNA degradation and protein turnover.

“We identified two nonsense mutants for TRPA1, one that possessed a 4bp insertion and one that possessed a 7bp deletion. […] In the TLR7 experiments, a pure TLR7^-/-^ line was established and compared to age-matched WT fish, since we were unable to genotype the 1bp mutation via conventional methods (gel electrophoresis, HRMA) and could only be identified via sequencing.

11) Is the IMQ-responsive subset of neurons in larvae genetically specified or determined stochastically? This could be addressed by imaging responses to IMQ at different times and determining if the same neurons are activated each time.

The reviewers raise an excellent point. We have performed the suggested experiments to image responses to IMQ at different times and determine whether the same neurons are activated each time. We found that nearly all of IMQ responsive neurons respond to multiple pulses of IMQ, arguing that this is a defined population of neurons and that IMQ responses are not determined stochastically. We have modified both the Results and Discussion sections of the paper to include this new information, and have added an additional figure (Figure 4—figure supplement 2).

Results: “To verify that IMQ activates a selective subset of TRPA1+ neurons as opposed to activating TRPA1+ neurons stochastically, we performed calcium imaging experiments in which 3dpf *elavl3*:H2BGCaMP6 were exposed to successive pulses of 100 μM IMQ. […] The finding that nearly all IMQ-responsive neurons were dual responders, argues that these neurons comprise a distinct population of TRPA1 expressing neurons, primed to respond to low intensity TRPA1 dependent stimuli.”

Discussion: “Our observation that sequential pulses of IMQ (100 μM) activate the low intensity TRPA1 agonist responsive population in zebrafish demonstrates that this population is likely genetically defined, and not determined stochastically.”

12) These experiments show that IMQ responses require TrpA1b, but not TLR7. They also show that in heterologous cells, expression of TrpA1b, but not TLR7, confers a response to IMQ, albeit more weakly than AITC. These results lead the authors to propose that TrpA1b is the direct sensor of IMQ, rather than a channel downstream of a GPCR. The fact that TrpA1b can be directly activated is compelling, but the results do not exclude the possibility that there is another receptor in vivo. This issue should be discussed.

We thank the reviewer for their thoughtful suggestion of discussing this issue more thoroughly, and have provided an additional paragraph in the Discussion section exploring the possibility of IMQ-evoked itch mediated by an unknown co-receptor. While the existence of an alternate IMQ-binding co-receptor is possible, we feel that this is improbable. First, results from our in vitro experiments necessitate that such a receptor would be endogenously expressed in HEK cells in order to couple with TRPA1 and evoke calcium flux, as we did not observe any response to IMQ in cells that were not transfected with TRPA1. This seems unlikely. Secondly, results from experiments in which we modified the stimulus intensity similarly make this possibility unlikely. The overlap we observed in the number of neurons that were responsive to IMQ and low-dose AITC similarly implies that the same receptor on these neurons was activated under these two conditions. If a neuron has two distinct ways of being activated by IMQ (either directly through TRPA1 or through the coupling of TRPA1 to a mystery co-receptor), we would not expect the neuronal recruitment to scale with stimulus intensity the way we observed.

Additionally, we have also elaborated upon the possibility of “indirect itch” that could occur when pruriceptors are activated indirectly as a result of an immune response generated via activation of TLR7 and/or other immune receptors. We think that these mechanisms most certainly exist, and are likely responsible for our observations that IMQ-induced scratching behavior is not entirely abolished in TRPA1^-/-^ animals, and that IMQ-induced scratching behavior is identical between TRPA1^-/-^ mice and their wildtype siblings at higher concentrations of IMQ (also discussed in our response to Essential revision #8; the indirect role of TLRs in fish responses are also discussed in our response to Essential revision #4). At such high concentrations, mechanisms involving the direct activation of TRPA1 to evoke itch may be saturated, and additional increases in responses may result from the indirect stimulation via immune modulation. We hope that this addition adequately addresses the reviewer’s concerns.

“Our data implicates a mechanism for the direct activation of TRPA1 by IMQ to elicit sensations of itch in both zebrafish and mice. […] Furthermore, the dose-dependent overlap we observed in the number of neurons that were responsive to IMQ and AITC, as well as similarities in the dose dependent itch and nociceptive behaviors elicited by these compounds in both zebrafish and mice, implies that these compounds are acting via the direct activation of TRPA1.”

13) The fact that low concentrations of AITC activate a similar fraction of TrpA1b-expressing neurons in zebrafish larvae as IMQ is a nice finding supporting the authors' hypothesis that IMQ-expressing neurons are primed for TrpA1 activation. This idea was strengthened by an experiment showing that IMQ-responsive DRG neurons are enriched in the population of neurons activated by low AITC concentrations. This finding could be strengthened further by addressing the same question with calcium imaging in zebrafish larvae- i.e. are the same trigeminal neurons that respond to IMQ also those responsive to low TrpA1 concentrations?

We realize that we may not have been explicit as necessary in our original text, but we did perform imaging experiments in which we sequentially applied IMQ, a low concentration of AITC (10 μM), and a high concentration of AITC (50 μM) to 3dpf *elavl3*:GCaMP5 zebrafish larvae. The results of these experiments are shown in Figure 4—figure supplement 1C. We show that the population of IMQ+ neurons is entirely contained within the population of neurons responding to low AITC concentrations, which is itself within a broader population of neurons that are activated by high concentrations of AITC. From this data, it can be extrapolated that IMQ responding neurons are enriched within the population of neurons that respond to low AITC levels. In the original Discussion section of the paper, we only referenced data obtained from mouse experiments, and following the reviewer’s suggestions, we have expanded our treatment of this topic to include results from zebrafish experiments.

Results section: “in vivo GCaMP imaging bolstered our CaMPARI findings that stimulus intensity affects which subpopulations of TRPA1+ neurons are activated. […] Only a subset of neurons that responded to a high concentration (50 μM) of AITC responded to a lower concentration (10 μM) of AITC (11/42), and of those even fewer neurons responded to 100 μM IMQ (4/11) (Figure 4—figure supplement 1C).”

Discussion: “Additionally, in both species, IMQ (100 μM) responsive neurons were enriched within populations of neurons that responded to low concentrations of AITC; in other words, we observed a high degree of overlap in the populations of neurons that responded to low intensity TRPA1 agonists irrespective of stimulus identity. Together, this provides further evidence of the greater sensitivity of a population of itch-encoding neurons to TRPA1 agonists.”